# Understanding allergic multimorbidity within the non-eosinophilic interactome

Daniel Aguilar[1,2,3]*, Nathanael Lemonnier[4], Gerard H. Koppelman[5,6], Erik Melén[7], Baldo Oliva[8], Mariona Pinart[2], Stefano Guerra[2,9], Jean Bousquet[10,11], Josep M. Anto[2]

**1** Biomedical Research Networking Center in Hepatic and Digestive Diseases (CIBEREHD), Instituto de Salud Carlos III, Barcelona, Spain, **2** ISGlobal, Barcelona Institute for Global Health, Barcelona, Spain, **3** 6AM Data Mining, Barcelona, Spain, **4** Institute for Advanced Biosciences, Inserm U 1209 CNRS UMR 5309 Université Grenoble Alpes, Site Santé, Allée des Alpes, La Tronche, France, **5** University of Groningen, University Medical Center Groningen, Beatrix Children's Hospital, Department of Pediatric Pulmonology and Pediatric Allergology, Groningen, Netherlands, **6** University of Groningen, University Medical Center Groningen, GRIAC Research Institute, **7** Institute of Environmental Medicine, Karolinska Institutet, Stockholm, Sweden, **8** Structural Bioinformatics Group, Research Programme on Biomedical Informatics, Department of Experimental and Health Sciences, Universitat Pompeu Fabra, Barcelona, Spain, **9** Asthma and Airway Disease Research Center, University of Arizona, Tucson, Arizona, United States of America, **10** Hopital Arnaud de Villeneuve University Hospital, Montpellier, France, **11** Charité, Universitätsmedizin Berlin, Humboldt-Universität zu Berlin, and Berlin Institute of Health, Comprehensive Allergy Center, Department of Dermatology and Allergy, Berlin, Germany

* daniel.aguilar@ciberehd.org

**Data Availability Statement:** All relevant data are within the paper and its Supporting Information files.

## Abstract

### Background

The mechanisms explaining multimorbidity between asthma, dermatitis and rhinitis (allergic multimorbidity) are not well known. We investigated these mechanisms and their specificity in distinct cell types by means of an interactome-based analysis of expression data.

### Methods

Genes associated to the diseases were identified using data mining approaches, and their multimorbidity mechanisms in distinct cell types were characterized by means of an *in silico* analysis of the topology of the human interactome.

### Results

We characterized specific pathomechanisms for multimorbidities between asthma, dermatitis and rhinitis for distinct emergent non-eosinophilic cell types. We observed differential roles for cytokine signaling, TLR-mediated signaling and metabolic pathways for multimorbidities across distinct cell types. Furthermore, we also identified individual genes potentially associated to multimorbidity mechanisms.

### Conclusions

Our results support the existence of differentiated multimorbidity mechanisms between asthma, dermatitis and rhinitis at cell type level, as well as mechanisms common to distinct

**Funding:** This work was supported by Mechanisms of the Development of ALLergy (MeDALL), a collaborative project done within the EU under the Health Cooperation Work Programme of the Seventh Framework programme (grant agreement number 261357). EM is supported by grants from the European Research Council (n° 757919) and the Swedish Research Council. NL is a recipient of a postdoctoral fellowship from the French National Research Agency in the framework of the "Investissements d'avenir" program (ANR-15-IDEX-02). The funders had no role in study design, data collection and analysis, decision to publish, or preparation of the manuscript. 6AM Data Mining provided support in the form of a salary for DA, but did not have any role in the study design, data collection and analysis, decision to publish, or preparation of the manuscript. The specific roles of these authors are articulated in the 'author contributions' section.

**Competing interests:** I have read the journal's policy and the authors of this manuscript have the following competing interests: DA worked for 6AM Data Mining data science company as a bioinformatics consultant at the time of the study. This commercial affiliation does not alter our adherence PLOS ONE policies on sharing data and materials. The authors have declared that no competing interests exist.

**Abbreviations:** A, asthma; AD, multimorbidity between asthma and dermatitis; ADR, multimorbidity between asthma, dermatitis and rhinitis; AR, multimorbidity between asthma and rhinitis; D, dermatitis; DR, multimorbidity between dermatitis and rhinitis; MS, Multimorbidity Score; PS, Perturbation Score; R, rhinitis.

cell types. These results will help understanding the biology underlying allergic multimorbidity, assisting in the design of new clinical studies.

## Introduction

Mapping diseases onto molecular interaction networks (such as the protein-protein interaction network, also known as the *interactome*), has contributed to the elucidation of disease mechanisms and the identification of new disease-associated genes [1, 2]. Evidence suggests that disease-associated genes are not randomly distributed within the interactome, but instead they work coordinately forming connected communities linked to disease phenotypes [1, 3–5]. Furthermore, genes expressed in a particular tissue tend to form a well-localized subnetwork, and the partition of the complete interactome into tissue-specific subnetworks has important implications for the understanding of disease mechanisms [6]. Gene activity is often dependent on tissue context, and human diseases arise from the complex interplay of tissue and cell-lineage-specific processes [7, 8]. Disease-associated genes are usually tissue-specific and their interaction patterns with other genes change in diseased tissues as compared to healthy ones [9]. These observations make elucidating the context-specific role of genes in pathophysiological processes particularly challenging [10, 11]. Exploiting tissue-specific information has provided valuable clues on tissue-specific gene functions [12].

The computational analysis of tissue-specific cellular networks helps to understand the tissue-specific mechanisms of diseases, and how those mechanisms interplay with one another. Authors have long hypothesized that perturbations of cellular networks are key to many phenotypic and pathophenotypic outcomes [1, 4, 13–16]. Because of this, co-morbid and multi-morbid phenotypes are expected to share tissue-specific causative mechanisms [12, 13]. Studies have found that multimorbidity between metabolic diseases can be explained by shared cellular mechanisms [17], and that multimorbidities do not necessarily imply that the involved diseases are linked through shared genes [16, 18–20].

In a previous work, we uncovered significant patterns of network connectivity between the cellular networks associated to asthma (A), dermatitis (D) and rhinitis (R) [21], which supported the idea that A, D and R form a multimorbidity cluster due to shared genes [22, 23] and pathomechanisms [24–26]. While eosinophils have been singled out as prominent mediators in a number of inflammatory diseases [27–30] and multimorbidities [31–34], many other cell types (e.g. macrophages, monocytes/dendritic cells, lymphocytes), are involved in complex and heterogeneous diseases such as A, D and R [35–37]. Yet, a cell-type-based interactome analysis of the allergic multimorbidity has not been reported to the best of our knowledge. In this study, we use the interactome and expression data to investigate the mechanisms of multimorbidity between A, D and R at a cell-type-specific level, focusing on emergent non-eosinophilic allergy-mediating cell types across distinct tissues. Our results provide new insights could provide valuable information to improve prevention and treatment of these diseases.

## Methods

Methods are described in detail in S1 Text.

### Data sources

**Gene-disease associations.** We built the sets of genes associated to A, D and R by integrating data from four sources: (1) The Comparative Toxicogenomics Database [38], which

provides highly reliable gene-disease associations characterized through various experimental procedures combined with a process of expert curation of the literature and other databases (e.g. OMIM [39]). (2) The DisGeNet catalog, that contains curated gene-disease associations extracted from literature [40]. (3) UniProt-derived gene-disease associations, extracted from the *Involvement in disease* section of the Uniprot Knowledgebase [41]. (4) The Phenotype-Genotype Integrator database, that integrates information various NCBI genomic databases with association data from the National Human Genome Research Institute GWAS Catalog [42]. This is the only data source containing solely GWAS-derived gene associations [43]. Genes associated to a disease *d* (any of A, D or R) will be hereinafter referred to as *d*-associated genes.

**The interactome.** We built the functional interaction network (hereinafter called the *interactome* for brevity) by combining data from: (1) The Reactome Functional Interaction Network (v. 022717) [44], which includes not only protein-protein interactions but also gene expression interaction, metabolic interactions and signal transduction. (2) The STRING interaction network (v.10.5) [45].

**Cell-type-specific gene expression.** Gene expression levels were obtained from the human gene expression atlas available at ArrayExpress under accession number E-MTAB-62 [46]. This is a cell-type-wide compendium of high-quality microarray-derived expression data that has been previously used in other network-based analysis of gene expression [47–49] and has been incorporated into a number of biomedical software packages [50–52]. We filtered the data to remove redundancies and samples subjected to particular treatments or environmental factors (see S1 Text). We then centered and standardized the expression level of each gene as:

$$e_{g,c} = \frac{(E_{g,c} - M_g)}{MAD_g}$$

where $E_{g,c}$ is the expression level of the gene $g$ in cell type $c$, $M_g$ is the median expression level the gene $g$ across all cell types, and $MAD_g$ is the median absolute deviation of the expression levels of gene $g$ across all cell types. This made the expression levels comparable between genes [53, 54].

We defined a gene to be cell-type-specific if its absolute normalized expression level $e_{g,c}$ was at least 1.5 larger than the interquartile range (IQR) of its normalized expression across all cell types [6, 12, 55, 56]. Genes specific to a cell type $c$ (any of our cell types of interest) will be hereinafter referred to as $c$-specific genes.

## Cellular pathways

Cellular pathways were downloaded from Reactome database in the *UniProt2Reactome* format files [44]. Pathway-associated genes either without expression data or not present in the interactome were not considered. Disease-related cellular pathways (e.g. *Constitutive Signaling by Aberrant PI3K in Cancer*) were not considered. Reactome is a collection of pathways built in a hierarchical manner, where larger pathways are subdivided into smaller pathways with more specific functionalities. This implies a trade-off between the specificity in the representation of cellular functions and the average number of genes per pathway [57]. To minimize the overlap between pathways in order to avoid redundancies that could negatively affect our analysis [58], we calculated the pairwise overlap between pathways at distinct levels of the Reactome hierarchy using the Sorensen-Dice method [59–61]. If two pathways had an overlap of > 50% genes, the one with the lowest number of associated genes was removed from the set. We chose pathways of at depth 3 of the hierarchy because it provided a mean overlap < 1% while annotating 4,809 genes (this is 87,9% of the total genes annotated in the database, all levels

considered). Genes associated to a pathway *p* (any of our pathways of interest) will be hereinafter referred to as *p*-associated genes.

Pathway annotation in our previous study of A, D and R were extracted from BioCarta [62]. There is not a perfect equivalence between cellular pathways from BioCarta and Reactome databases, so in order to compare our results to those from our previous whole-organism multimorbidity study [21] we performed an association test to identify which BioCarta pathways significantly overlapped with Reactome pathways (Fisher's Exact test, adjusted *P* <0.05; S1 Table). *P*-values in this study were adjusted by the Benjamini-Hochberg method for false discovery (FDR) control [63].

## Cell-type-specific networks

In order to generate the specific network for any cell type *c*, we selected all edges from the interactome connecting *c*-specific genes [6, 64]. Because of the interactome-based nature of our analysis, those genes not present in the interactome or not present in the expression dataset were removed from the analysis. The statistical significance of the number of *d*-associated genes present in each *c*-specific network was calculated by means of a Fisher's Exact test (adjusted *P* <0.05).

## Quantifying cell-type-specific multimorbidity

In order to obtain a quantitative measure of the extent to which A, D and R multimorbidity is manifested in distinct cell types, we designed an interactome-based approach (workflow in Fig 1; illustrated with an example in S1 Fig). Briefly, we scored all genes specific to a given cell type according to their connectivity (or their "closeness") to known disease-associated genes, under the rationale that the malfunction of one (or more) of the disease-associated genes is likely to perturb the function of the neighboring genes, eventually disrupting a cellular mechanism and giving rise to a diseased phenotype [5, 65–68]. In other words, we scored each gene in each cell type according to its contribution to the manifestation of A, D and R. Then, we selected the set of top-scoring genes (called *S*; $S^c_d$ being the top-scoring genes for disease *d* in cell type *c*). Finally, for each cell type we calculated the overlap between the sets of top-scoring genes for AD, AR, DR and ADR. This overlap was called the Multimorbidity Score (*MS*; $MS^T_{d1,d2}$ being the Multimorbidity Score for diseases *d1* and *d2* in cell type *c*). The process is described in detail in S1 Text.

## Characterizing cell-type-specific multimorbidity mechanisms

After having quantitatively scored the multimorbidity between diseases in different cell types, we wished to identify the actual cellular mechanisms involved in the manifestation of the multimorbidities. To do so, we designed a method to measure the perturbation that a disease can exert over a cellular pathway in a given cell type. The starting point is the set of top-scoring genes $S^c_d$ calculated in the previous section. We identified the set of cellular pathways present in cell type *c*, and then scored how perturbed they were by the manifestation of disease *d* using $S^c_d$ (workflow in Fig 2; illustrated with an example in S2 Fig). This score was called the Perturbation Score (*PS*; $PS^c_{p,d}$ being the perturbation experimented by pathway *p* during the manifestation of disease *d* on cell type *c*). Under the assumption that any disease can be viewed as the product of perturbed cellular mechanisms (i.e. cellular pathways), and that multimorbidity is known to arise as those perturbed mechanisms are shared by distinct diseases [12, 13, 69, 70], we selected as candidate mechanisms for multimorbidity those pathways that were significantly perturbed in more than one disease in the same cell type. The process is described in detail in S1 Text.

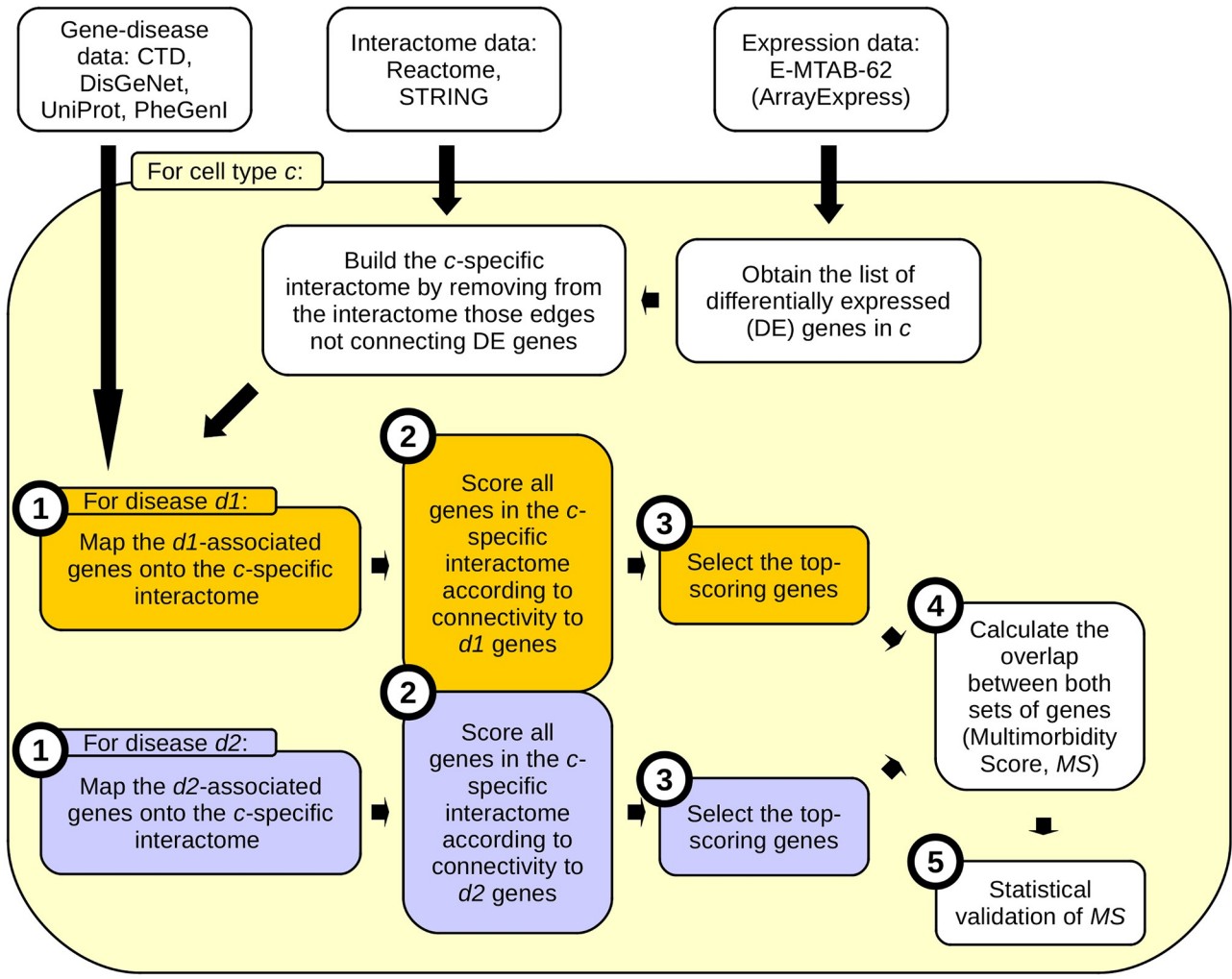

**Fig 1. Workflow for _Quantifying cell-type-specific multimorbidity_ section.** Only multimorbidity between two diseases is shown. Numbered circles indicate the steps of in the section _Quantifying cell-type-specific multimorbidity_ in _Methods_.

### Identifying cell-type-specific candidates to multimorbidity

Lastly, we wished to identify individual genes that might constitute candidates to multimorbidity. In the _Quantifying cell-type-specific multimorbidity_ section we had identified the sets of genes more susceptible to be perturbed by a disease in a cell type ($S^c_d$). We identified as multimorbidity candidates those genes simultaneously belonging to $> = 2$ of those sets (i.e. susceptible to be perturbed by two diseases in the same cell type) for AD, AR, DR multimorbidities, and $> = 3$ in the case of ADR multimorbidity. In addition, we numerically scored the contribution of each gene $g$ to multimorbidity ($MS^{g,c}_{d1,d2}$ being the Multimorbidity Score for gene $c$ with respect to diseases $d1$ and $d2$ in cell type $c$). This process is detailed in **Text S1**.

## Results

### Gene-disease associations

The number of genes associated to A, D and R with representation in the interactome and expression data was 98, 62 and 10, respectively. The complete list of genes is shown in Table 1

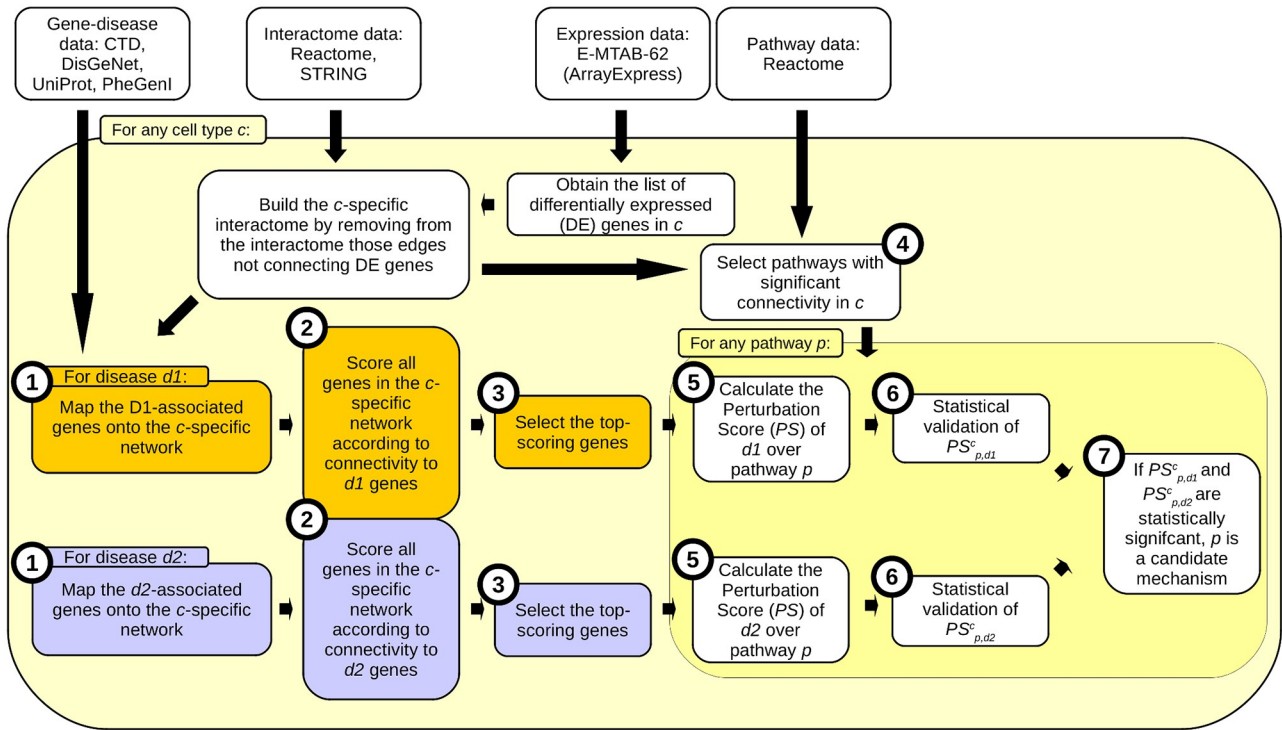

**Fig 2. Workflow for *Characterizing cell-type-specific multimorbidity mechanisms* section.** Only multimorbidity between two diseases is shown. Numbered circles indicate the steps of in the section *Characterizing cell-type-specific multimorbidity mechanisms* in *Methods*.

(see S2 Table and *Gene-disease associations* in the Methods section for data sources). Three genes were associated with A, D and R: *IL13*, platelet-activating factor acetylhydrolase *PLA2G7* and *LRRC32*, a signal peptide cleavage essential for surface expression of a regulatory T cell surface protein. The complete list of all disease-associated genes in each cell type is provided in S2 Table.

## Cell-type-specific gene expression and the cell-type-specific networks

The complete interactome contained 15,332 genes (nodes) and 394,317 interactions (edges). The total number of cell types was 60, classified into 15 distinct tissues. The total number of genes with expression data was 8,461 (of which 7,486 were present in the interactome). Table 2 shows the number of genes specific to each cell-type-specific network and its statistical significance (an extended version of the table with *p*-values is provided as S3 Table). The number of genes present in a cell-type-specific network is lower than the number of cell-type-specific genes because we only considered directly connected cell-type-specific gene pairs. In other words, for a cell type *c*, any *c*-specific gene not connected to other *c*-specific gene was not a part of the *c*-specific network. The cell type with the most specific genes was *hematopoietic stem cell* with 1,156 specific genes. The cell type with the least specific genes was blood-derived *monocyte* with 132 genes. The complete list of tissues, cell types and cell-type-specific genes is available at S2 Table.

## Cellular pathways

The number of pathways in Reactome database was 519 after filtering, with an average pairwise overlap of 0.01%. Overall, 6,989 genes were associated to at least one pathway. On average,

**Table 1. Gene-disease associations.**

| gene name | A | D | R | gene description | gene name | A | D | R | gene description |
|---|---|---|---|---|---|---|---|---|---|
| IL13 | ● | ● | ● | interleukin 13 | MMP9 | ● | | | matrix metallopeptidase 9 |
| LRRC32 | ○ | ○ | ○ | leucine rich repeat containing 32 | MS4A2 | ● | | | membrane spanning 4-domains A2 |
| PLA2G7 | ● | ● | ● | phospholipase A2 group VII | MYB | ● | | | MYB proto-oncogene, transcription factor |
| CASP8 | ● | ● | | caspase 8 | NDFIP1 | ○ | | | Nedd4 family interacting protein 1 |
| CCL11 | ● | ● | | C-C motif chemokine ligand 11 | NFKB2 | ● | | | nuclear factor kappa B subunit 2 |
| CD14 | ● | ● | | CD14 molecule | NOS2 | ● | | | nitric oxide synthase 2 |
| CHI3L1 | ● | ● | | chitinase 3 like 1 | NPY | ● | | | neuropeptide Y |
| CRNN | ○ | ○ | | cornulin | PARP1 | ● | | | poly(ADP-ribose) polymerase 1 |
| EFHC1 | ○ | ○ | | EF-hand domain containing 1 | PEX14 | ○ | | | peroxisomal biogenesis factor 14 |
| ETS1 | ○ | ○ | | ETS proto-oncogene 1, transcription factor | PHF11 | ● | | | PHD finger protein 11 |
| IL18R1 | ○ | ○ | | interleukin 18 receptor 1 | PLAU | ● | | | plasminogen activator, urokinase |
| IL1B | ● | ● | | interleukin 1 beta | PPP2CA | ● | | | protein phosphatase 2 catalytic subunit alpha |
| IL33 | ● | ● | | interleukin 33 | PTEN | ● | | | phosphatase and tensin homolog |
| IL4 | ● | ● | | interleukin 4 | PTGES | ○ | | | prostaglandin E synthase |
| IL5 | ● | ● | | interleukin 5 | PTGS2 | ● | | | prostaglandin-endoperoxide synthase 2 |
| IL6R | ○ | ○ | | interleukin 6 receptor | RNASE3 | ● | | | ribonuclease A family member 3 |
| IRAK3 | ● | ● | | interleukin 1 receptor associated kinase 3 | RORA | ○ | | | RAR related orphan receptor A |
| KIF3A | ● | ○ | | kinesin family member 3A | SCGB1A1 | ● | | | secretoglobin family 1A member 1 |
| RAD50 | ● | ○ | | RAD50 double strand break repair protein | SOD1 | ● | | | superoxide dismutase 1 |
| SPINK5 | ● | ● | | serine peptidase inhibitor, Kazal type 5 | TBX21 | ● | | | T-box 21 |
| STAT6 | ○ | ● | | signal transducer and activator of transcription 6 | TBXA2R | ● | | | thromboxane A2 receptor |
| TNIP1 | ● | ○ | | TNFAIP3 interacting protein 1 | TGFB1 | ● | | | transforming growth factor beta 1 |
| IL1RL1 | ● | | ○ | interleukin 1 receptor like 1 | TIMP3 | ● | | | TIMP metallopeptidase inhibitor 3 |
| RANBP6 | ○ | | ○ | RAN binding protein 6 | TNC | ● | | | tenascin C |
| SLC25A46 | ○ | | ○ | solute carrier family 25 member 46 | TNFSF4 | ○ | | | TNF superfamily member 4 |
| SMAD3 | ○ | | ○ | SMAD family member 3 | TRPA1 | ● | | | transient receptor potential cation channel subfamily A member 1 |
| TLR1 | ○ | | ○ | toll like receptor 1 | TYRP1 | ○ | | | tyrosinase related protein 1 |
| ADCYAP1R1 | ● | | | ADCYAP receptor type I | VEGFA | ● | | | vascular endothelial growth factor A |
| ADORA1 | ○ | | | adenosine A1 receptor | CCL17 | | ● | | C-C motif chemokine ligand 17 |
| ALDH2 | ● | | | aldehyde dehydrogenase 2 family (mitochondrial) | CCL22 | | ● | | C-C motif chemokine ligand 22 |
| ALOX5 | ● | | | arachidonate 5-lipoxygenase | CCL24 | | ● | | C-C motif chemokine ligand 24 |
| AREG | ● | | | amphiregulin | CCR5 | | ● | | C-C motif chemokine receptor 5 (gene/pseudogene) |
| ARG1 | ● | | | arginase 1 | CD207 | | ○ | | CD207 molecule |
| ARG2 | ● | | | arginase 2 | CSTA | | ● | | cystatin A |
| BACH2 | ○ | | | BTB domain and CNC homolog 2 | CTLA4 | | ● | | cytotoxic T-lymphocyte associated protein 4 |
| BCL2 | ● | | | BCL2, apoptosis regulator | CXCL10 | | ● | | C-X-C motif chemokine ligand 10 |
| CAT | ● | | | catalase | CYP24A1 | | ○ | | cytochrome P450 family 24 subfamily A member 1 |
| CCL2 | ● | | | C-C motif chemokine ligand 2 | EMSY | | ● | | EMSY, BRCA2 interacting transcriptional repressor |
| CDH17 | ○ | | | cadherin 17 | FOXP3 | | ● | | forkhead box P3 |
| CDK2 | ○ | | | cyclin dependent kinase 2 | GLB1 | | ● | | galactosidase beta 1 |
| CFTR | ● | | | cystic fibrosis transmembrane conductance regulator | IFNG | | ● | | interferon gamma |
| CHIT1 | ○ | | | chitinase 1 | IL10 | | ● | | interleukin 10 |
| CPN1 | ● | | | carboxypeptidase N subunit 1 | IL15RA | | ○ | | interleukin 15 receptor subunit alpha |
| CRB1 | ○ | | | crumbs 1, cell polarity complex component | IL18RAP | | ○ | | interleukin 18 receptor accessory protein |
| CRBN | ○ | | | cereblon | IL2RA | | ● | | interleukin 2 receptor subunit alpha |
| CXCL14 | ● | | | C-X-C motif chemokine ligand 14 | IL6 | | ● | | interleukin 6 |

*(Continued)*

**Table 1.** (Continued)

| gene name | A | D | R | gene description | gene name | A | D | R | gene description |
|---|---|---|---|---|---|---|---|---|---|
| CYSLTR2 | ● | | | cysteinyl leukotriene receptor 2 | IL7R | | | ○ | interleukin 7 receptor |
| DNMT1 | ● | | | DNA methyltransferase 1 | KRT1 | | ● | | keratin 1 |
| EDN1 | ● | | | endothelin 1 | PAH | | ● | | phenylalanine hydroxylase |
| ELF3 | ○ | | | E74 like ETS transcription factor 3 | PFDN4 | | | ○ | prefoldin subunit 4 |
| GPR37L1 | ○ | | | G protein-coupled receptor 37 like 1 | PPP2R3C | | | ○ | protein phosphatase 2 regulatory subunit B"gamma |
| GRM4 | ○ | | | glutamate metabotropic receptor 4 | PTPRN2 | | | ○ | protein tyrosine phosphatase, receptor type N2 |
| GSDMB | ● | | | gasdermin B | REL | | | ○ | REL proto-oncogene, NF-kB subunit |
| GSTM1 | ● | | | glutathione S-transferase mu 1 | RTEL1-TNFRSF6B | | | ○ | RTEL1-TNFRSF6B readthrough (NMD candidate) |
| GSTP1 | ● | | | glutathione S-transferase pi 1 | S100A8 | | ● | | S100 calcium binding protein A8 |
| HERC2 | ○ | | | HECT and RLD domain containing E3 ubiquitin protein ligase 2 | SELE | | ● | | selectin E |
| HMOX1 | ● | | | heme oxygenase 1 | SLC11A1 | | ● | | solute carrier family 11 member 1 |
| HNMT | ● | | | histamine N-methyltransferase | SPRR1B | | | ○ | small proline rich protein 1B |
| HTATIP2 | ○ | | | HIV-1 Tat interactive protein 2 | SPRR3 | | | ○ | small proline rich protein 3 |
| ICAM1 | ● | | | intercellular adhesion molecule 1 | STAT1 | | ● | | signal transducer and activator of transcription 1 |
| IKZF3 | ● | | | IKAROS family zinc finger 3 | TGM5 | | ● | | transglutaminase 5 |
| IL12B | ● | | | interleukin 12B | TNFRSF1B | | ● | | TNF receptor superfamily member 1B |
| IL1RL2 | ○ | | | interleukin 1 receptor like 2 | TNXB | | | ○ | tenascin XB |
| IL1RN | ● | | | interleukin 1 receptor antagonist | VAX2 | | | ○ | ventral anterior homeobox 2 |
| IL2RB | ○ | | | interleukin 2 receptor subunit beta | VNN1 | | ● | | vanin 1 |
| KRT19 | ● | | | keratin 19 | VNN2 | | ● | | vanin 2 |
| LPP | ○ | | | LIM domain containing preferred translocation partner in lipoma | WAS | | ● | | Wiskott-Aldrich syndrome |
| MLLT3 | ○ | | | MLLT3, super elongation complex subunit | WIPF1 | | ● | | WAS/WASL interacting protein family member 1 |
| MMP10 | ● | | | matrix metallopeptidase 10 | BDH1 | | | ○ | 3-hydroxybutyrate dehydrogenase 1 |
| MMP13 | ○ | | | matrix metallopeptidase 13 | FOXJ1 | | | ● | forkhead box J1 |

A: asthma; D: dermatitis; R: rhinitis. Filled circle: all evidences. Empty circle: GWAS-only evidence. Only genes with expression data, present in the interactome and associated to A, D or R are shown.

~37% of genes on cell-type-specific networks were associated to at least one pathway. The fraction of pathway-associated genes present in each cell type is shown in S4 Table. The list of genes associated to each pathway in each cell-type-specific network is provided in S5 Table. The connectivity $C^c_p$ of the pathways is shown in S6 Table. As an example, Fig 3 shows cellular pathway (*Regulation of TLR by endogenous ligand*) mapped onto a cell-type specific network (CD19+ B cell).

## Quantification of cell-type-specific multimorbidity

The Multimorbidity Score (*MS*) quantitatively measured the multimorbidity between A, D and R specific to different cell types (Table 3). S2 Table contains the number of top-scoring genes for each disease on each cell-type-specific network ($|S^c_d|$, see Methods). Of the 60 cell-type-specific networks, 12 were associated to a single disease and were not considered for further multimorbidity analysis. Inspection of Table 3 shows 14 cell types associated to ADR multimorbidity because their *MS* value is > 0 for all combinations of the three diseases (the strength of the association given by the MS value, ranging from 0 to 1). The cell types include monocytes-macrophages, T cells and plasma cells, as well as skin endothelial cells and esophageal epithelial cells. These 14 cell types will be subject to scrutiny in the following sections.

**Table 2. Number of disease-associated genes on cell-type-specific networks.**

| tissue | cell type | Cell-type-specific genes | Cell-type-specific network genes | | | | | | |
|---|---|---|---|---|---|---|---|---|---|
| | | n | n | A | | D | | R | |
| | | | | n | % | n | % | n | % |
| Adipose tissue from abdomen | Adipose-derived adult stem cells (ADASCs) | 584 | 319 | 9 | 2.8 | 7 | 2.2 | 1 | 0.3 |
| Adipose tissue from abdomen and thigh | Adipose-derived adult stem cells (ADASCs) | 645 | 343 | 8 | 2.3 | 5 | 1.5 | | |
| Aorta | Primary aortic smooth muscle cell | 1023 | 623 | 7 | 1.1 | 3 | 0.5 | | |
| Blood | 721 B lymphoblasts | 534 | 329 | 8 | 2.4 | 2 | 0.6 | | |
| | BDCA4+ dentritic cell | 719 | 386 | 12 | 3.1 | 4 | 1 | | |
| | CD14+ monocyte | 786 | 426 | 13 | 3.1 | 8 | 1.9 | 1 | 0.2 |
| | CD19+ B cell (neg. sel.) | 1027 | 619 | 11 | 1.8 | 16 | 2.6 | 1 | 0.2 |
| | CD34+ cell | 433 | 295 | 7 | 2.4 | 2 | 0.7 | | |
| | CD34+ hematopoietic stem cell | 941 | 639 | 6 | 0.9 | 3 | 0.5 | | |
| | CD34+ T cell | 219 | 91 | 4 | 4.4 | 4 | 4.4 | | |
| | CD4+ T cell | 622 | 315 | 10 | 3.2 | 6 | 1.9 | 1 | 0.3 |
| | CD8+ T cell | 344 | 131 | 4 | 3.1 | 1 | 0.8 | | |
| | Central memory 1 CD4+ T cell | 228 | 72 | 1 | 1.4 | 3 | 4.2 | | |
| | Central memory CD4+ T cell | 220 | 79 | 3 | 3.8 | 6 | 7.6 | | |
| | Effector memory CD4+ T cell | 154 | 47 | 4 | 8.5 | 7 | 14.9 | | |
| | Erythrocyte | 247 | 140 | 8 | 5.7 | 9 | 6.4 | 1 | 0.7 |
| | Granulocyte | 342 | 203 | 3 | 1.5 | 2 | 1 | | |
| | Hematopoietic stem cell | 1148 | 723 | 11 | 1.5 | 3 | 0.4 | | |
| | Lymphocyte | 348 | 255 | 11 | 4.3 | 14 | 5.5 | 1 | 0.4 |
| | Macrophage | 382 | 225 | 11 | 4.9 | 14 | 6.2 | 2 | 0.9 |
| | Monocyte | 147 | 91 | 8 | 8.8 | 8 | 8.8 | 1 | 1.1 |
| | Monocyte derived macrophage | 430 | 266 | 10 | 3.8 | 14 | 5.3 | 2 | 0.8 |
| | Naive CD4+ T cell | 248 | 89 | 3 | 3.4 | 1 | 1.1 | | |
| | Primary bone marrow CD34+ stem cell | 398 | 199 | 5 | 2.5 | 3 | 1.5 | 1 | 0.5 |
| | Progenitor cell, hematopoietic stem cell | 440 | 207 | 6 | 2.9 | 2 | 1 | | |
| | T cell | 532 | 284 | 16 | 5.6 | 13 | 4.6 | 1 | 0.4 |
| | T lymphocyte | 193 | 88 | 5 | 5.7 | 5 | 5.7 | | |
| Bone marrow | CD138+ plasma cell | 936 | 526 | 15 | 2.9 | 9 | 1.7 | 3 | 0.6 |
| | Immature-B cell | 212 | 87 | 1 | 1.1 | | | | |
| | Mesenchymal stem cell | 307 | 175 | 4 | 2.3 | 1 | 0.6 | | |
| | Mesenchymal stem cell BM-MSC | 474 | 263 | 4 | 1.5 | | | | |
| | Pre-B-I cell | 444 | 229 | 4 | 1.7 | | | | |
| | Pre-B-II large cell | 830 | 485 | 3 | 0.6 | | | | |
| | Pre-B-II small cell | 466 | 242 | 3 | 1.2 | 1 | 0.4 | | |
| | Primary bone marrow CD34- mesenchymal stem cell | 209 | 80 | 3 | 3.8 | | | | |
| | Primary bone marrow CD34+ stem cell | 252 | 116 | 10 | 8.6 | 3 | 2.6 | | |
| Connective tissue | Fibroblast | 305 | 131 | 4 | 3.1 | 3 | 2.3 | | |
| Esophagus | Esophageal epithelium | 702 | 399 | 11 | 2.8 | 10 | 2.5 | 1 | 0.3 |
| Eye | Trabecular meshwork | 540 | 319 | 4 | 1.3 | | | | |
| | Trabecular meshwork cell | 561 | 297 | 6 | 2 | 1 | 0.3 | | |
| Kidney | Epithelium | 596 | 346 | 6 | 1.7 | 3 | 0.9 | | |
| Mesagnium | Mesangial cell | 396 | 169 | 3 | 1.8 | | | | |
| Ovary | Theca | 746 | 393 | 3 | 0.8 | | | | |

*(Continued)*

**Table 2.** (Continued)

| | | Cell-type-specific genes | Cell-type-specific network genes | | | | | | |
| | | n | n | A | | D | | R | |
| tissue | cell type | | | n | % | n | % | n | % |
|---|---|---|---|---|---|---|---|---|---|
| **Palatine tonsil** | CXCR5(-)ICOS(-/lo) CD4+ T cell | 196 | 83 | 2 | 2.4 | | | | |
| | CXCR5(hi)ICOS(hi) CD4+ T cell | 212 | 66 | 4 | 6.1 | 4 | 6.1 | | |
| | CXCR5(lo)ICOS(int) CD4+ T cell | 174 | 55 | 2 | 3.6 | 3 | 5.5 | | |
| **Skin** | Epidermis and dermis | 650 | 385 | 12 | 3.1 | 10 | 2.6 | 1 | 0.3 |
| | Primary blood vessel endothelial cell | 314 | 138 | 6 | 4.3 | 1 | 0.7 | 1 | 0.7 |
| | Primary lymphatic endothelial cell | 348 | 152 | 4 | 2.6 | 1 | 0.7 | | |
| | Primary microvascular endothelial cell | 446 | 213 | 6 | 2.8 | 1 | 0.5 | 1 | 0.5 |
| **Skin (leg)** | Epidermis and dermis | 658 | 378 | 12 | 3.2 | 10 | 2.6 | | |
| **Thymus** | CD34+CD1a- thymocyte | 334 | 139 | 1 | 0.7 | 1 | 0.7 | | |
| | CD34+CD38- thymocyte | 801 | 479 | 3 | 0.6 | 1 | 0.2 | | |
| | DP CD3- thymocyte | 319 | 170 | 1 | 0.6 | | | | |
| | DP CD3+ thymocyte | 402 | 140 | 1 | 0.7 | 2 | 1.4 | | |
| | ISP CD4+ thymocyte | 340 | 200 | 1 | 0.5 | | | | |
| | SP CD4+ thymocyte | 346 | 121 | 1 | 0.8 | 1 | 0.8 | | |
| | SP CD8+ thymocyte | 270 | 90 | 1 | 1.1 | | | | |
| | Thyrocyte | 406 | 218 | 7 | 3.2 | 4 | 1.8 | | |
| **Uterine tube** | Primary uterine smooth muscle cell | 460 | 227 | 6 | 2.6 | 3 | 1.3 | | |

A: asthma. D: dermatitis. R: rhinitis. Light blue background: the number of genes is significantly higher than random expectation (adjusted $P < 0.05$). Dark blue background: the number of genes is significantly higher than random expectation (adjusted $P < 0.01$). For clarity, zero values are represented as blank cells, and cell types without any disease-associated genes are not shown.

S7 Table provides a combined overview of the results of Tables 2 and 3, containing the cell types with a significant number of A-, D- or R-associated genes as well as those cell types with nonzero *MS*.

## Cell-type-specific multimorbidity mechanisms

Table 4 shows the pathways identified as candidate mechanisms for multimorbidity in the 14 cell types where *MS* for ADR is >0 (Table 3), where pathways in the *Cytokine signaling in immune system* category roughly correspond to the pathways activated in the type-2 asthmatic response (particularly, *IL4* and *IL13* signaling [71, 72]). S8 Table shows candidate mechanisms in all other cell types (which are restricted to AD multimorbidity except for one pathway in primary bone marrow CD34+ stem cells, associated to AR multimorbidity). It is noteworthy that some cell types do not present any significant mechanism for multimorbidity despite being associated to multimorbidity in Table 3 (namely, epidermis/dermis, and primary microvascular endothelial cells, not associated to any pathway). Other cell types are strongly associated to ADR multimorbidity while not being associated to any mechanism for ADR multimorbidity. This is the case of CD14+ monocytes, for which only a mechanism mediation AD multimorbidity (*NOD1/2 signaling pathway*) was found. The reason for these observations is that, on average, only ~37% of genes in a given cell type are annotated to a least one pathway (S4 Table). Thus, a large number of non-annotated genes might be still contributing to multimorbidity. The cellular pathways perturbed in each individual disease and cell type (i.e $PS^c_{pd}$ significant at $P < 0.05$, see Methods) are provided in S9 Table.

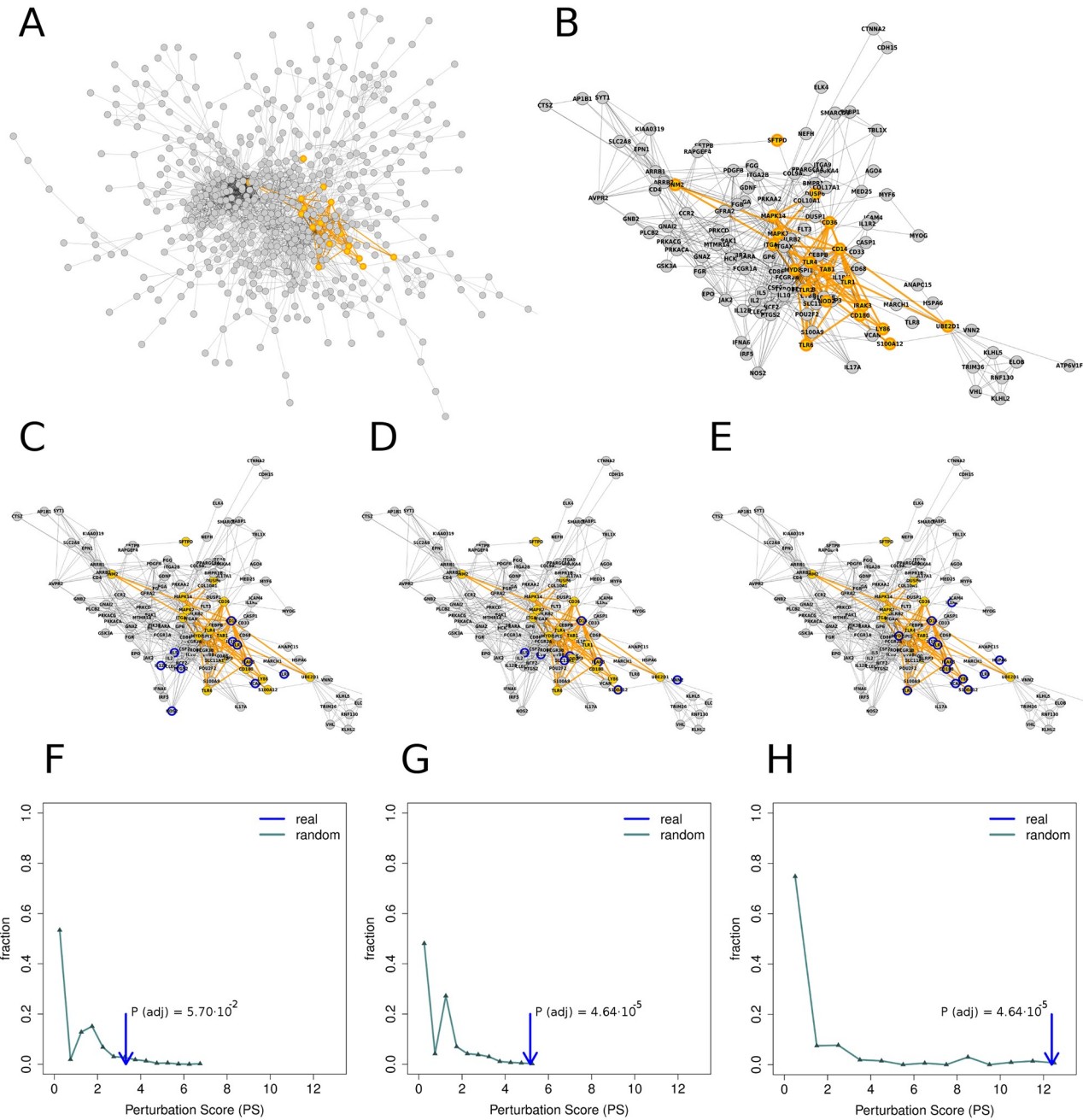

**Fig 3. Pathway *Toll Like Receptor 4 TLR4 Cascade* on the *CD19+ B cell* specific network. (A)** Complete view of the largest component of the network. Pathway-associated genes and their interactions are shown in orange. **(B)** Zoom to the pathway-associated genes and their closest neighbors only. Pathway-associated genes and their connections are shown in orange. **(C)** Top-scoring asthma genes (see Methods) are shown with blue borders. **(D)** Top-scoring dermatitis genes are shown with blue borders. **(E)** Top-scoring rhinitis genes are shown with blue borders. The fraction of pathway genes within the top-scoring gene sets is only significant for dermatitis and rhinitis. **(F-H)** Distribution of random Perturbation Score (*PS*) for A, D and R, respectively. An arrow represents the real *PS*. Pathways whose *PS* is significantly larger than random expectation (*P* < 0.05, panels G and H) are denoted as perturbed in the respective disease.

## Candidate multimorbidity genes

Table 5 shows the 30 top-scoring candidate genes for multimorbidity (and S10 Table contains the full collection of candidate genes). The score assigned to multimorbidity (columns AD,

**Table 3. Cell-type-specific multimorbidities between asthma, dermatitis and rhinitis.**

| tissue | cell type / line | AD | AR | DR | ADR |
|---|---|---|---|---|---|
| Adipose tissue from abdomen and thigh | Adipose-derived adult stem cells (ADASCs) | 0.35 | | | |
| Adipose tissue from abdomen | Adipose-derived adult stem cells (ADASCs) | 0.35 | 0.25 | 0.36 | 0.24 |
| Aorta | Primary aortic smooth muscle cell | 0.29 | | | |
| Blood | 721 B lymphoblasts | 0.08 | | | |
| | BDCA4+ dentritic cell | 0.18 | | | |
| | CD14+ monocyte | 0.77 | 0.71 | 0.83 | 0.70 |
| | CD19+ B cell (neg. sel.) | 0.17 | 0.33 | 0.21 | 0.09 |
| | CD34+ T cell | 0.12 | | | |
| | CD34+ cell | 0.65 | | | |
| | CD34+ hematopoietic stem cell | 0.20 | | | |
| | CD4+ T cell | 0.58 | 0.50 | 0.58 | 0.31 |
| | CD8+ T cell | 0.11 | | | |
| | Central memory 1 CD4+ T cell | 0.57 | | | |
| | Central memory CD4+ T cell | 0.33 | | | |
| | Effector memory CD4+ T cell | 0.36 | | | |
| | Erythrocyte | 0.38 | 0.38 | 0.22 | 0.24 |
| | Granulocyte | 0.33 | | | |
| | Hematopoietic stem cell | 0.19 | | | |
| | Lymphocyte | 0.42 | 0.33 | 0.37 | 0.24 |
| | Macrophage | 0.17 | 0.29 | 0.24 | 0.11 |
| | Monocyte | 0.38 | 0.50 | 0.33 | 0.30 |
| | Monocyte derived macrophage | 0.15 | 0.24 | 0.21 | 0.10 |
| | Naive CD4+ T cell | 0.50 | | | |
| | Primary bone marrow CD34+ stem cell | | 0.45 | | |
| | Progenitor cell, hematopoietic stem cell | 0.22 | | | |
| | T cell | 0.48 | 0.15 | 0.16 | 0.11 |
| | T lymphocyte | 0.40 | | | |
| Bone marrow | CD138+ plasma cell | 0.38 | 0.47 | 0.71 | 0.33 |
| | Pre-B-II small cell | 0.07 | | | |
| | Primary bone marrow CD34+ stem cell | 0.29 | | | |
| Connective tissue | Fibroblast | 0.14 | | | |
| Esophagus | Esophageal epithelium | 0.27 | 0.43 | 0.33 | 0.29 |
| Kidney | Epithelium | 0.11 | | | |
| Palatine tonsil | CXCR5(hi)ICOS(hi) CD4+ T cell | 0.50 | | | |
| | CXCR5(lo)ICOS(int) CD4+ T cell | 0.40 | | | |
| Skin (leg) | Epidermis and dermis | 0.26 | | | |
| Skin | Epidermis and dermis | 0.35 | 0.14 | 0.16 | 0.13 |
| | Primary blood vessel endothelial cell | 0.38 | 0.11 | 0.20 | |
| | Primary lymphatic endothelial cell | 0.12 | | | |
| | Primary microvascular endothelial cell | 0.50 | 0.50 | 1.00 | 0.60 |
| Thyroid | Thyrocyte | 0.54 | | | |
| Uterine tube | Primary uterine smooth muscle cell | 0.11 | | | |

The gradient of red correspond to the values of the Multimorbidity Score (MS, indicated within the cells; $0 \leq MS \leq 1$). Empty cells have a MS = 0.

**Table 4. Cellular pathways associated to multimorbidity between asthma, dermatitis and rhinitis.**

| category | pathway | Adipose tissue from abdomen — Adipose-derived adult stem cells (ADASCs) | Blood — CD14+ monocyte | Blood — CD19 + B cell (neg. sel.) | Blood — CD4 + T cell | Blood — Erythrocyte | Blood — Lymphocyte | Blood — Macrophage | Blood — Monocyte | Blood — Monocyte derived macrophage | Blood — T cell | Bone marrow — CD138+ plasma cell | Esophagus — Esophageal epithelium | Skin — Epidermis and dermis | Skin — Primary microvascular endothelial cell |
|---|---|---|---|---|---|---|---|---|---|---|---|---|---|---|---|
| Metabolism of carbohydrates | Heparan sulfate heparin HS-GAG metabolism | | | AR | | | | | | | | | | | |
| | Chondroitin sulfate dermatan sulfate metabolism | | | AR | | | | | | | | | | | |
| Apoptosis | Ligand-dependent caspase activation | | | | AD | | | | | | ADR | | | | |
| Signaling by GPCR | G-protein beta gamma signalling | | | | | | ADR | | | | | | | | |
| Death receptor signalling | TNFR1-induced proapoptotic signaling | | | | AD | | | | | | | | | | |
| Cytokine signaling in immune system | Interleukin-1 signaling | | | ADR | | | ADR | | | | AD | | AR | | |
| | Other interleukin signaling | | | | | | | | | | | | | | |
| | Interleukin-10 signaling | AD | | ADR | DR | | AD | AD | | | AD | ADR | | | |
| | Interleukin-4 and 13 signaling | AD | | | | | AD | | | | AD | | | | |
| Adaptive immune system | Antigen processing-Cross presentation | | | DR | | AR | | | | | | | | | |
| Innate immune system | Toll Like Receptor 4 TLR4 Cascade | | | ADR | AD | AR | | | AR | | | ADR | | | |
| | Toll Like Receptor 9 TLR9 Cascade | | | ADR | | | | | | | | | | | |
| | Toll Like Receptor 10 TLR10 Cascade | | | DR | | | | | | | | | | | |
| | Toll Like Receptor 3 TLR3 Cascade | | | DR | | | | | | | | | | | |
| | Toll Like Receptor 2 TLR2 Cascade | | | ADR | | AR | | | | AR | | ADR | | | |
| | Regulation of TLR by endogenous ligand | | | DR | | AR | | | AR | | DR | AD | | | |
| | NOD1/2 Signaling Pathway | | AD | | | | | | | | | | | | |

Red cells: multimorbidity between A and D. Orange cells: multimorbidity between A and R. Light blue cells: multimorbidity between D and R. Dark blue cells: multimorbidity between A, D and R. Only cell types with MS > 0 for multimorbidity between A, D and R are shown.

**Table 5. Candidate genes associated to multimorbidity between A, D and R.**

| tissue | cell type | gene | AD | AR | DR | ADR | A | D | R | DAP12 signaling | Interleukin-1 signaling | Interleukin-17 signaling | Other interleukin signaling | Interleukin-2 signaling | Interleukin-3, 5 and GM-CSF signaling | Interleukin-6 family signaling | Interleukin-10 signaling | Interleukin-4 and 13 signaling | Interleukin-20 family signaling | Interferon alpha beta signaling |
|---|---|---|---|---|---|---|---|---|---|---|---|---|---|---|---|---|---|---|---|---|
| Skin | Primary microvascular endothelial cell | IL1RL1 | 8.09 | 8.09 | 10.25 | 8.81 | • | | • | | | | • | | | | | | | |
| Skin | Primary microvascular endothelial cell | IL33 | 8.09 | 8.09 | 10.25 | 8.81 | • | • | | | | | • | | | | | | | |
| Esophagus | Esophageal epithelium | IL13 | 5.62 | 9.89 | 9.16 | 8.22 | • | • | • | | | | | | | • | • | | |
| Skin | Epidermis and dermis | PLA2G7 | 4.71 | 9.68 | 9.29 | 7.90 | • | • | • | | | | | | | | | | |
| Adipose tissue from abdomen | Adipose-derived adult stem cells (ADASCs) | IL33 | 5.53 | 8.72 | 8.95 | 7.73 | • | • | | | | | • | | | | | | | |
| Adipose tissue from abdomen | Adipose-derived adult stem cells (ADASCs) | IL1RL1 | 4.77 | 9.31 | 8.07 | 7.38 | • | | • | | | | • | | | | | | | |
| Blood | CD14+ monocyte | IL13 | 5.78 | 7.21 | 7.67 | 6.88 | • | • | • | | | | | | | • | • | | |
| Esophagus | Esophageal epithelium | IL33 | 6.63 | 7.32 | 6.48 | 6.81 | • | • | | | | | • | | | | | | | |
| Bone marrow | CD138+ plasma cell | PLA2G7 | 5.67 | 6.84 | 5.89 | 6.13 | • | • | • | | | | | | | | | | |
| Blood | CD4+ T cell | IL13 | 4.95 | 6.53 | 6.79 | 6.09 | • | • | • | | | | | | | • | • | | |
| Bone marrow | CD138+ plasma cell | IL13 | 5.16 | 5.92 | 6.41 | 5.83 | • | • | • | | | | | | | • | • | | |
| Bone marrow | CD138+ plasma cell | TLR1 | 3.96 | 7.07 | 5.51 | 5.51 | • | | • | | | | | | | | | | |
| Bone marrow | CD138+ plasma cell | CD14 | 6.07 | 4.76 | 5.48 | 5.44 | • | • | | | | | | | | | | | |
| Esophagus | Esophageal epithelium | IL22RA1 | 3.45 | 6.63 | 6.20 | 5.43 | | | | | | | | | | | | • | |
| Blood | CD14+ monocyte | IL18R1 | 6.19 | 4.83 | 5.21 | 5.41 | • | • | • | | | | • | | | | | | |
| Skin | Epidermis and dermis | BCHE | 2.37 | 6.80 | 6.66 | 5.28 | • | | | | | | | | | | | | |
| Blood | CD14+ monocyte | IL5 | 5.91 | 4.39 | 4.92 | 5.07 | • | • | • | | • | • | • | | | | | | |
| Blood | CD19+ B cell (neg. sel.) | IRAK3 | 5.33 | 4.23 | 4.10 | 4.55 | • | • | | • | | | | | | | | | |
| Esophagus | Esophageal epithelium | IL20RA | 2.92 | 5.24 | 4.84 | 4.33 | • | • | | | | | | | | | | • | |
| Blood | CD14+ monocyte | ARG1 | 4.28 | 5.17 | 3.33 | 4.26 | • | | | | | | | | | | | | |
| Blood | CD14+ monocyte | IL18RAP | 4.46 | 3.10 | 5.21 | 4.26 | | • | | | | | | | | | | | |
| Blood | CD14+ monocyte | IL11 | 3.25 | 4.32 | 4.75 | 4.10 | | | | | | | | • | | | | | |
| Blood | CD14+ monocyte | IFNA8 | 3.25 | 4.32 | 4.75 | 4.10 | | | | | | | | | | | | | • |
| Blood | CD19+ B cell (neg. sel.) | CD14 | 5.05 | 3.57 | 3.48 | 4.03 | • | • | | | | | | | | | | | |

*(Continued)*

**Table 5.** (Continued)

| tissue | cell type | gene | AD | AR | DR | ADR | A | D | R | Antigen processing-Cross presentation | Defensins | Toll Like Receptor 4 TLR4 Cascade | Toll Like Receptor 9 TLR9 Cascade | Toll Like Receptor 3 TLR3 Cascade | Toll Like Receptor 7 8 TLR7 8 Cascade | Toll Like Receptor 2 TLR2 Cascade | FCERI mediated MAPK activation | Regulation of TLR by endogenous ligand | TRAF6 mediated IRF7 activation |
|---|---|---|---|---|---|---|---|---|---|---|---|---|---|---|---|---|---|---|---|
| Bone marrow | CD138+ plasma cell | RNASE3 | 4.00 | 4.90 | 2.94 | 3.95 | • | | | | | | | | | | | | |
| Blood | CD14+ monocyte | FOXP3 | 4.17 | 2.68 | 4.95 | 3.93 | | • | | | | | | | | | | | |
| Blood | Monocyte derived macrophage | IL13 | 3.51 | 4.03 | 3.81 | 3.78 | • | • | • | | | | | | | | • | • | |
| Blood | Lymphocyte | IL13 | 3.34 | 3.98 | 3.78 | 3.70 | • | • | • | | | | | | | | • | • | |
| Blood | Macrophage | IL13 | 3.30 | 3.85 | 3.71 | 3.62 | | • | • | | | | | | | | • | • | |
| Blood | CD14+ monocyte | IL9 | 2.84 | 3.77 | 4.06 | 3.56 | | | | | | | • | | | | | | | |
| Skin | Primary microvascular endothelial cell | IL1RL1 | 8.09 | 8.09 | 10.25 | 8.81 | • | | • | • | | | | | | | | | |
| Skin | Primary microvascular endothelial cell | IL33 | 8.09 | 8.09 | 10.25 | 8.81 | • | • | • | | | | | | | | | | |
| Esophagus | Esophageal epithelium | IL13 | 5.62 | 9.89 | 9.16 | 8.22 | • | • | • | | | | | | | | | | |
| Skin | Epidermis and dermis | PLA2G7 | 4.71 | 9.68 | 9.29 | 7.90 | • | • | • | | | | | | | | | | |
| Adipose tissue from abdomen | Adipose-derived adult stem cells (ADASCs) | IL33 | 5.53 | 8.72 | 8.95 | 7.73 | • | • | | | | | | | | | | | |
| Adipose tissue from abdomen | Adipose-derived adult stem cells (ADASCs) | IL1RL1 | 4.77 | 9.31 | 8.07 | 7.38 | • | | • | | | | | | | | | | |
| Blood | CD14+ monocyte | IL13 | 5.78 | 7.21 | 7.67 | 6.88 | • | • | • | | | | | | | | | | |
| Esophagus | Esophageal epithelium | IL33 | 6.63 | 7.32 | 6.48 | 6.81 | • | • | • | | | | | | | | | | |
| Bone marrow | CD138+ plasma cell | PLA2G7 | 5.67 | 6.84 | 5.89 | 6.13 | • | • | • | | | | | | | | | | |
| Blood | CD4+ T cell | IL13 | 4.95 | 6.53 | 6.79 | 6.09 | • | • | • | | | | | | | | | | |
| Bone marrow | CD138+ plasma cell | IL13 | 5.16 | 5.92 | 6.41 | 5.83 | • | • | • | | | | | | | | | | |
| Bone marrow | CD138+ plasma cell | TLR1 | 3.96 | 7.07 | 5.51 | 5.51 | • | • | • | • | • | • | | | | • | | • | |
| Bone marrow | CD138+ plasma cell | CD14 | 6.07 | 4.76 | 5.48 | 5.44 | • | • | • | • | • | • | • | • | • | • | | • | |
| Esophagus | Esophageal epithelium | IL22RA1 | 3.45 | 6.63 | 6.20 | 5.43 | • | • | | | | | | | | | | | |
| Blood | CD14+ monocyte | IL18R1 | 6.19 | 4.83 | 5.21 | 5.41 | • | • | | | | | | | | | | | |
| Skin | Epidermis and dermis | BCHE | 2.37 | 6.80 | 6.66 | 5.28 | | | | | | | | | | | • | | |
| Blood | CD14+ monocyte | IL5 | 5.91 | 4.39 | 4.92 | 5.07 | • | • | | | • | | | | | | • | | |
| Blood | CD19+ B cell (neg. sel.) | IRAK3 | 5.33 | 4.23 | 4.10 | 4.55 | • | • | | | • | | | | | • | | | |
| Esophagus | Esophageal epithelium | IL20RA | 2.92 | 5.24 | 4.84 | 4.33 | | | | | | | | | | | | | |

(Continued)

**Table 5.** (Continued)

| Tissue | Cell type | Gene | AD | AR | DR | ADR | | | | | | |
|---|---|---|---|---|---|---|---|---|---|---|---|---|
| Blood | CD14+ monocyte | ARG1 | 4.28 | 5.17 | 3.33 | 4.26 | • | | | | | |
| Blood | CD14+ monocyte | IL18RAP | 4.46 | 3.10 | 5.21 | 4.26 | • | | | | | |
| Blood | CD14+ monocyte | IL11 | 3.25 | 4.32 | 4.75 | 4.10 | | | | | | |
| Blood | CD14+ monocyte | IFNA8 | 3.25 | 4.32 | 4.75 | 4.10 | | | | | | • |
| Blood | CD19+ B cell (neg. sel.) | CD14 | 5.05 | 3.57 | 3.48 | 4.03 | • | • | • | • | • | • |
| Bone marrow | CD138+ plasma cell | RNASE3 | 4.00 | 4.90 | 2.94 | 3.95 | • | | | | | |
| Blood | CD14+ monocyte | FOXP3 | 4.17 | 2.68 | 4.95 | 3.93 | • | | | | | |
| Blood | Monocyte derived macrophage | IL13 | 3.51 | 4.03 | 3.81 | 3.78 | • • | • | | | | |
| Blood | Lymphocyte | IL13 | 3.34 | 3.98 | 3.78 | 3.70 | • | • | | | | |
| Blood | Macrophage | IL13 | 3.30 | 3.85 | 3.71 | 3.62 | • | • | | | | |
| Blood | CD14+ monocyte | IL9 | 2.84 | 3.77 | 4.06 | 3.56 | | | | | | |

Column AD (red background): score of the gene in multimorbidity between A and D. Column AR (orange background): score of the gene in multimorbidity between A and R. Column DR (light blue background): score of the gene in multimorbidity between D and R. Column ADR (dark blue background): multimorbidity between A, D and R. Scores within columns AD to ADR are the average z-scores for each gene in each cell type for the corresponding diseases (see Methods). Column A: a dot indicates that the gene is known to be associated to asthma. Column D: a dot indicates that the gene is known to be associated to dermatitis. Column R: a dot indicates that the gene is known to be associated to rhinitis. Columns labeled after pathways: a dot indicates that the gene is known to be associated to the corresponding pathway (for brevity, only pathways related to the immune system are shown). Genes in the table are ranked according to their average score across the AD, AR, DR and ADR columns, and only the 30 top-scoring genes are shown.

AR, DR, ADR in Table 5 and S10 Table) can be read as the importance of the gene as mediator for multimorbidity. As expected, many of the top-scoring candidates are associated to immune system pathways. It is noteworthy that some genes may be associated to pathways which are, in fact, not characterized as multimorbidity mechanisms. For instance, Table 5 shows *IL13* gene as a strong ADR multimorbidity candidate in esophageal epithelium. This gene is annotated as belonging to the *Interleukin-10 signaling* an *Interleukin-4 and 13 signaling* pathways. However, neither pathway was characterized as a mechanism of multimorbidity for esophageal epithelium in Table 4, because their perturbation score $PS^T_{pd}$ did not reach statistical significance. Genes in Table 5 show a higher score, on average, for AR than for AD multimorbidity ($P = 0.01482$; paired Wilcoxon-Mann-Whitney test), implying a more closely-knit biological mechanism for AR than for AD multimorbidity. The same was observed for AD vs DR ($P = 1.02 \cdot 10^{-3}$; paired Wilcoxon-Mann-Whitney test) but not for AR vs DR. This observation was also true when comparing scores of the whole set of predicted genes in S10 Table. Comparisons are shown in S11 Table. Genes which are not known be associated to any of the diseases under study (i.e. they are not present in Table 1) but were characterized as candidates for multimorbidity are particularly interesting candidates for experimental characterization. There are 100 genes of this kind, and 21 of them are candidates for ADR multimorbidity. Table 6 shows the 30 top-scoring ones.

## Discussion

In this study, we have performed an interactome-based analysis of expression data to characterize specific mechanisms for multimorbidity between asthma (A), dermatitis (D) and rhinitis (R) in distinct 14 non-eosinophilic cell types and 15 tissues. We observed differential roles for cytokine signaling, particularly associated with type 2 inflammation, TLR-mediated signaling and metabolic pathways for multimorbidities across distinct cell types. Furthermore, we also identified individual genes potentially associated to multimorbidity mechanisms.

### Strengths

Interactome-based computational analysis provide a global view of the increasing complexity of disease-gene association data, and the relationships among diseases, genes and functions [73]. By employing an expression compendium that incorporates information on multiple heterogeneous gene expression experiments, we were able to identify cell-type-specific mechanisms that underlie the multimorbidity between A, D and R, focusing on 14 cell types that are emerging as major components in these complex diseases in 15 distinct tissues. Although eosinophils are an important cell type in A [30, 74], we focused on other important yet no so well-studied cell types in connection to ADR multimorbidity.

Our approach characterizes the mechanisms of multimorbidity not only by analyzing the contributions of individual genes, but also their interrelationship and their connectivity to other genes within the interactome. This is relevant because molecular causes of multimorbidity are not restricted to shared genes, but involve a cascade of common perturbed cellular mechanisms without which the whole mechanisms of multimorbidity cannot be properly characterized. Although the statistical analysis of the overlap between sets of genes has been widely employed to uncover disease-disease and disease-pathway associations, the limited knowledge of disease-associated genes and lack of annotation data have hampered its results [75, 76]. More recent approaches incorporating interactome-derived data provided a substantial improvement to characterize multimorbidity [20, 65, 76, 77]. Our approach can detect multimorbidity even if no shared genes are involved by identifying the cell-type-specific mechanisms associated to multimorbidity. In this respect, and because cellular pathways represent a

**Table 6. Candidate genes associated to multimorbidity between A, D and R, and not associated to any of the diseases.**

| tissue | cell type | gene | AD | AR | DR | ADR | Interleukin-1 signaling | Interleukin-12 family signaling | Other interleukin signaling | Interleukin-6 family signaling | Interleukin-10 signaling | Interleukin-4 and 13 signaling | Interleukin-20 family signaling | Interferon alpha beta signaling | Antigen processing-Cross presentation | ZBP1DAI mediated induction of type I IFNs | Toll Like Receptor 4 TLR4 Cascade | Toll Like Receptor 2 TLR2 Cascade | Regulation of innate immune responses to cytosolic DNA | Regulation of TLR by endogenous ligand | Inflammasomes | TRAF6 mediated IRF7 activation | Negative regulators of RIG-1 MDA5 signaling |
|---|---|---|---|---|---|---|---|---|---|---|---|---|---|---|---|---|---|---|---|---|---|---|---|
| Esophagus | Esophageal epithelium | IL22RA1 | 3.45 | 6.63 | 6.20 | 5.43 | | | | | | | • | | | | | | | | | | |
| Skin | Epidermis and dermis | BCHE | 2.37 | 6.80 | 6.66 | 5.28 | | | | | | | | | | | | | | | | | |
| Blood | CD19+ B cell (neg. sel.) | NCAN | | 5.93 | | | | | | | | | | | | | | | | | | | |
| Blood | CD19+ B cell (neg. sel.) | CSPG5 | | 5.93 | | | | | | | | | | | | | | | | | | | |
| Esophagus | Esophageal epithelium | IL20RA | 2.92 | 5.24 | 4.84 | 4.33 | | | | | | | • | | | | | | | | | | |
| Blood | CD14+ monocyte | IL1 | 3.25 | 4.32 | 4.75 | 4.10 | | | | • | | | | | | | | | | | | | |
| Blood | CD14+ monocyte | IFNA8 | 3.25 | 4.32 | 4.75 | 4.10 | | | | | | | | • | | | | | | | | • | |
| Blood | CD14+ monocyte | IL9 | 2.84 | 3.77 | 4.06 | 3.56 | | | • | | | | | | | | | | | | | | |
| Bone marrow | CD138+ plasma cell | CD180 | 3.17 | 3.60 | 3.88 | 3.55 | | | | | | | | | | | • | | | | | | |
| Bone marrow | CD138+ plasma cell | RNASE2 | 2.78 | 4.03 | 3.76 | 3.52 | | | | | | | | | | | | | | | | | |
| Bone marrow | CD138+ plasma cell | EPX | 2.78 | 4.03 | 3.76 | 3.52 | | | | | | | | | | | | | | | | | |
| Blood | CD14+ monocyte | PRLR | 2.90 | 3.66 | 3.97 | 3.51 | | | | | | | | | | | | | | | | | |
| Blood | Granulocyte | NLRP3 | 4.67 | | | | | | | | | | | | | | | | | | • | | |
| Blood | CD19+ B cell (neg. sel.) | VCAN | | 3.97 | | | | | | | | | | | | | | | | | | | |
| Blood | CD4+ T cell | IL22 | | | 3.63 | | • | | | | | | • | | | | | | | | | | |
| Blood | CD14+ monocyte | IL23A | 2.32 | 2.86 | 3.05 | 2.75 | | • | | | | | | | | | | | | | | | |
| Blood | CD4+ T cell | IL1RA | | | 3.35 | | | | | | | | | | | | | | | | | | |
| Blood | CD4+ T cell | PRLR | | | 3.35 | | | | | | | | | | | | | | | | | | |
| Bone marrow | CD138+ plasma cell | TLR6 | 2.11 | 2.86 | 3.13 | 2.71 | | | | | | | | | • | | • | • | | • | | | |
| Blood | CD14+ monocyte | HPCAL4 | 2.35 | 2.95 | 2.64 | 2.64 | | | | | | | | | | | | | | | | | |
| Blood | CD14+ monocyte | CHP2 | 2.35 | 2.95 | 2.64 | 2.64 | | | | | | | | | | | | | | | | | |
| Blood | CD14+ monocyte | CIB2 | 2.35 | 2.95 | 2.64 | 2.64 | | | | | | | | | | | | | | | | | |
| Blood | CD14+ monocyte | OCM2 | 2.35 | 2.95 | 2.64 | 2.64 | | | | | | | | | | | | | | | | | |
| Bone marrow | CD138+ plasma cell | LBP | 2.26 | 2.56 | 2.62 | 2.48 | | | | | | | | | | | • | | | • | | | |
| Adipose tissue from abdomen and thigh | Adipose-derived adult stem cells (ADASCs) | MTMR8 | 3.69 | | | | | | | | | | | | | | | | | | | | |
| Esophagus | Esophageal epithelium | IL18 | | 2.98 | | | | | • | | • | • | | | | | | | | | | | |
| Blood | CD4+ T cell | CHP2 | | 2.91 | | | | | | | | | | | | | | | | | | | |
| Blood | CD4+ T cell | ZBP1 | 2.54 | 2.11 | 2.50 | 2.38 | | | | | | | | | | • | | | • | | | | |
| Blood | CD4+ T cell | RNF216 | 2.54 | 2.11 | 2.50 | 2.38 | | | | | | | | | | | | | | | | | • |
| Skin | Primary blood vessel endothelial cell | CCNA1 | | | 2.77 | | | | | | | | | | | | | | | | | | |

Column contents and background colors are as in Table 5. Genes in the table are ranked according to their average score across the AD, AR, DR and ADR columns, and only the 30 top-scoring genes are shown.

curated set of gene functions which may be only partially present in some cell types, our method allows not only to statistically quantify if a pathway can be considered as a specific multimorbidity mechanism in a cell type, but also the discovery of particular genes involved in the multimorbidity process. Finally, our method is fully scalable approach, making it possible to study and characterize the etiology critical for multimorbidity between large groups of diseases. The findings of this *in silico* study are hypothesis-generating and are intended to guide new experiments on cell-type-specific allergic multimorbidity. Consequently, they should be confirmed by proper mechanistic and genetic studies.

## Weaknesses

As usual in differential expression studies, we are considering the gene expression level as a proxy for the gene activity. However, these two characteristics do not always match. For instance, a gene can be significantly over-expressed in a certain tissue or cell type and yet, at the same time, its product can be rendered inactive through a post-translational modification (e.g. phosphorylation). Our methodology does not capture those cases. Similarly, the time-dependent gene expression patterns are not captured in our study, which only considers an interactome static in time.

Lack of data availability also limited our analysis. Eosinophils are not a part of the expression compendium used in this study. However, to the best of our knowledge, no cell-type- or tissue-wide expression compendium resolving eosinophils as an individual cell type exists. This is why we chose to focus our attention in other cell types, important yet no so well-studied in connection to ADR multimorbidity. Furthermore, our dataset reflects only expression levels in healthy individuals because no cell-type-wide expression compendium in subjects with ADR multimorbidity exists.

Another limitation of our study is data completeness. The intersection of expression and interactome data sources yields a low coverage of the complete genome. Although this is a common limitation (and authors have argued that the current coverage of the human interactome does not limit its successful application to the investigation of disease mechanisms [5, 16]), some data loss is unavoidable: for instance, a protein such as filaggrin (*FLG*), commonly associated to multimorbidity between A and D [78], was not present in our expression dataset and could not be incorporated to the study. Also, our expression dataset contains data primarily from adult subjects. Thus, it is unclear if our results can be generalized to other age groups like young children or elderly people. However, we believe that gain in knowledge largely compensates these limitations.

As for disease-gene associations, we are including gene-disease associations partially derived from GWAS studies, whose reliability has been questioned [79–81]. Additionally, the current human interactome is highly biased toward highly studied genes (a category that includes many disease-associated genes), representing only a very small densely connected fraction of the full interactome [82–86]. This bias might be larger than expected and may have an impact on the biological conclusions extracted from the studies of the interactome [87]. However, non-biased interactomes have a much lower coverage, which makes them unsuitable for some topology-based studies [87]. We tried to address this effect by building null models which take into account the degree of the original genes. It should be also noted that there are numerous factors, other than genetic ones, that determine multimorbidity, some of which are environmental, lifestyle-related or treatment-induced. Finally, different mutations on the same gene can have different pathological effects on its gene products [88]. We considered all disease-associated mutations to have an effect on gene activity that, in turn, has a molecular impact on the interactome.

## Quantification of cell-type-specific multimorbidity

The *MS* measure treats multimorbidity symmetrically with respect to the diseases being compared, meaning that it numerically reflects the mutual influence that the manifestation of one disease exerts over the other disease in a cell type. It can be interpreted as a measure of the degree to which a multimorbidity is present and specific to a certain cell type (regardless the fact that systemic mechanisms may be playing a role in multimorbidity as well, a case which is not captured by our method). Lower *MS* values imply that the specific mechanisms of the diseases are largely detached from each other in the corresponding cell type: the perturbation caused by the manifestation of one disease *d1* will be less likely to travel throughout the network and perturb the mechanisms that give rise to disease *d2*. At *MS* = 0 there is no multimorbidity between the diseases in the corresponding cell type (although multimorbidity may be present as a more systemic process). At *MS* = 1, the mechanisms of both diseases are identical in that cell type. We find an example of this in the primary microvascular endothelial cells, where *MS* = 1 for the DR multimorbidity. The implication of this is that not only the gene sets associated to D and R are identical in this cell type, but also that the gene sets influenced (or perturbed) by the malfunction of those genes are also identical, thus rendering both diseases the same disease in mechanistic terms for this cell type. Our methodology identifies cell-type-specific interactomes that are not exclusive of a single cell type: some parts of the interactome can be shared by more two or more (usually related) cell types.

*MS* revealed that all cell types with a significant number of disease-associated genes in at least one disease also display some degree of multimorbidity. For instance, genes associated to A and D are significantly associated to the monocyte cell-type-specific network (Table 2), which also displays a *MS* > 0 across all multimorbidities (AD, AR, DR, ADR; Table 3). The reverse, however, is not necessarily true: primary microvascular endothelial cells displayed high *MS* values despite not showing any significant gene association. The reason lies in the use of interactome data, which takes into account the interconnectivity amongst genes as well as their number, allowing for the identification of multimorbidities that would go unnoticed in a standard association analysis. In this line, it is also of note that a significant number of disease-associated genes in a cell type does not necessarily imply a stronger *MS*. For instance, macrophages and monocyte-derived macrophages have a significant number of disease-associated genes for A and D, and yet their *MS* value for AD multimorbidity are 0.17 and 0.15, respectively. As another example, CD14+ T cells show large *MS* values for all multimorbidities despite the fact that no statistical association was found neither with D- nor with R-associated genes in this cell type.

## Cell-type-specific multimorbidity mechanisms

Cytokine signaling, critical to the induction of the type 2 response, seems to be the main mechanism behind AD multimorbidity, and it is present in a number of distinct cell types, blood-related or not (Table 4, S8 Table). *IL4* and *IL13* have long been known to be amongst the cytokines secreted by Th2 cells in response to allergen-induced IgE synthesis in A, and the existence of an underlying *IL4*- and *IL13*-mediated pathomechanism for this multimorbidity has been suggested by a number of observations, for instance the response to similar treatments (e.g. dupilumab, a human monoclonal antibody that inhibits this type of signaling) [3]. *IL10*-associated signaling, a regulator of other proinflammatory cytokines [89], was also found as a contributing mechanism for AD multimorbidity across many cell types, as was *IL1*-associated signaling. *IL1* is a known inflammatory marker associated to D and bronchial A [90], amongst other diseases with inflammatory components. Interestingly, a role for *IL1* as a mediator in multimorbidities has already been hinted, as *IL1* blocking therapies have proven effective

against conditions encountered as comorbidities in patients with rheumatic diseases [91, 92]. We have to point out, however, that the definition of a pathway (as a functionally annotated gene set) should be taken into account when analyzing those results. For instance, the pathway *Antigen Processing and Cross-Presentation* is associated to AR multimorbidity in erythrocytes (Table 4). This contradicts evidence on MHC presence in human nonnucleated cells [93] because of the definition of the pathway in Reactome database, that includes genes also annotated in TLR-mediated pathways.

On the other hand, innate immune response mediated by toll-like receptors (TLRs) seems to be the key mechanism for multimorbidities implicating R. The TLR family of genes is important in barrier homeostasis and in the activation of the innate immune system [94], and there are evidences of its involvement in R [95–97]. Although the link between A and R is well established (the "United Airways" concept [98, 99]), there is limited knowledge about the mechanistic interplay between A and R [100, 101]. AR multimorbidity seems also largely restricted to a few blood-related cell types: CD19+ B cells, monocytes and erythrocytes. Genetic studies have linked the *TLR6-TLR1* locus to a role in the development of R [102], and changes in *TLR1* have been reported in asthmatic patients [17, 103], but no direct association between A and R is known. Similarly, changes in *TLR2* and *TLR4* expression are known to disturb the skin barrier in D [104]. According to our observations, *TLR4*-mediated cascade might play an important role in R-associated multimorbidities in blood-related cell types.

Esophageal epithelium cells seem to be also associated to AR multimorbidity by means of *IL1* signaling pathway and genes such as *IL-13* and *IL-33*. It is known that chronic eosinophilic inflammation of the esophagus is associated with tissue remodeling and fibrosis that shares many traits with A [105, 106]. Patients with eosinophilic esophagitis often present multimorbid conditions that include A and D [107]. It is also noteworthy the role of metabolism of proteoglycans in CD19+ B cells for this multimorbidity. In this sense, our results indicate that structurally similar proteoglycans neurocan (*NCAN*) and versican (*VCAN*) are related to this mechanism. Although no evidence linking these two genes to A or R is known, *VCAN* encodes an extracellular matrix protein that has been associated with A in murine models and with bronchiolitis in humans [108, 109].

We observed that cells of the skin epidermis/dermis, and primary microvascular endothelial cells were not significantly associated to any pathway. A number of reasons explain this observation: first, as already noted in the *Results* section, annotated pathways only cover approximately one-third of all genes in our cell-type-specific networks, leaving room for yet-unannotated mechanisms to play a critical role in multimorbidity. Second, our approach identifies *significantly* perturbed pathways, implying that some pathways may be perturbed without reaching the statistical significance cutoff of α = 0.05. Finally, our study only reflects cell-type-specific mechanisms, not excluding the existence of systemic mechanisms that may have relevant impact in a number of cell types. The fact that no pathway was characterized for these cell types, however, does not preclude the existence of individual candidate genes which might be playing a role in multimorbidity in them (see next section).

## Cell-type-specific candidate genes

We identified a number of individual genes as potentially associated to multimorbidity (Tables 5 and 6; S10 Table). The identification of candidate genes complements the characterization of mechanisms of multimorbidity based on pathway annotation. For instance, interleukin 1 receptor-like 1 (*IL1RL1*) is amongst the top-scoring candidates for ADR multimorbidity in primary microvascular endothelial skin-derived cells, yet it is not associated to any pathway in this cell type (and, thus, its contribution would have been lost had we focused solely on

pathway-annotated multimorbidity mechanisms). A candidate gene in a particular cell type may belong to a pathway not identified as a mechanism in that cell type. This is the case, for example, of the *IL13* gene, a high-scoring candidate gene in esophageal epithelium for ADR multimorbidity. This gene belongs to two pathways: *Interleukin-4 and 13 signaling* and *Interleukin-10 signaling*, and yet none of the two pathways is identified as a significant mechanism for this cell type and multimorbidity (because when considering all the pathway-associated genes, neither pathways is found to be significantly perturbed). Thus, we can conclude that *IL13* may play and important role as a multimorbidity mediator. Our results also provide valuable information of cell-type-specificity of candidate genes. For instance, *IL4* and *IL5*, two of the main inflammatory cytokines, in are associated to monocytes but not to macrophages (S10 Table), in agreement with previous observations [110].

The only non-cytokine-related gene in the top 10 positions of Table 5 is the *PLA2G7* gene, which controls inflammation though the inactivation of platelet-activating factor (*PAF*), a potent phospholipid-derived mediator of inflammation that is secreted by many immune cells and controls vascular permeability. Although no study associating *PLA2G7* to ADR multimorbidity exists, it is a strong candidate if we take into account the wide range of actions of *PAF* (considered a universal biological regulator [111]) that in turn associates *PLA2G7* to a number of inflammatory conditions other than A, D or R [112–114]. The "United Airways" concept, introduced in the previous section, is also supported by our results: on average, mechanisms mediating between A and R are more closely-knit (represented by higher average scores) than mechanisms mediating A and D, although A and D share more disease-associated genes (Table 1).

Some of the highest-scoring candidate genes were not even associated to any of the diseases of interest (Table 6) illustrating the potential of our approach to characterize yet-undescribed molecular mechanisms of multimorbidity. We predict interleukin receptors *IL22RA1* and *IL20RA* to play an important role in the ADR multimorbidity in the esophageal epithelium. To date, *IL22RA1* had been only associated to inflammatory responses in airway epithelia by genetic studies, and *IL20RA* to psoriasis [115, 116]. Also, the functional nature of genes in Table 6 is also much more diverse than that of Table 5. This is strongly suggestive of a research bias towards already-known cytokine-related mediators when it comes to the study of these allergic diseases, overlooking other functional groups. For instance, the second highest-scoring gene in Table 6 is butyrylcholinesterase *BCHE*, a poorly-studied detoxifying enzyme that has been proposed as a marker to identify and prognose systemic inflammation [117, 118] and that has only marginally associated to allergic diseases. *BCHE* is highlighted by our method as a mediator in ADR multimorbidity in skin. Table 6 also shows that the role of proteoglycans seems to be restricted to AR multimorbidity only through neurocan (*NCAN*) and chondroitin sulfate proteoglycan 5 (*CSPG5*). Although proteoglycans are known to influence the remodelling of nasal mucosa in R [119], no evidence exists linking them to allergic multimorbidity. However, our results indicate that TLRs are characteristically associated to multimorbidity involving R (Table 4), so there may be an interesting link between TLRs and proteoglycans in relation to AR multimorbidity, since it is known that chondroitin sulfate proteoglycans have the ability to bind TLRs and activating macrophages [120].

## Comparison to our previous study

In our previous *in silico* study of multimorbidity between A, D and R, we explored multimorbidity at whole organism level [21]. In this study we incorporated additional data that reflects the spatial cell-type-specific nature of the diseases and their multimorbidity. This presents a key opportunity to better understand the mechanisms of diseases, since cell-type-specific data

provides a more accurate picture of multimorbidity. We incorporated changes in the methodology as well. It remains focused on exploiting the topology of the interactome, but adopting a more complex approach that measures the role of pathways not only in terms of their direct interactions to individual disease-associated genes, but in terms of their global connectivity to those genes within a specific network.

Methodologically, differences in the gene-disease data sources used in both studies have an impact in the characterization of disease-associated genes. Also, availability of expression data limited the number of genes present in the study. For instance, thymic stromal lymphopoietin (*TSLP*), found to be associated to A, D and R in our previous study, is absent in this study because it was not present in the expression compendium. Also, pathway annotation in our previous study was extracted from BioCarta database, which is no longer updated, which made us chose Reactome database instead.

One of the main findings in our previous study was the significant role of eosinophilic-mediated pathways in AD multimorbidity (BioCarta pathways *CCR3 signaling in Eosinophils* and *The Role of Eosinophils in the Chemokine Network of Allergy* were identified with a high score). Because eosinophils were not included in the present study, the Reactome equivalents of those pathways (S1 Table) are not present in our results, confirming that our observations can be linked to mechanisms mediated by other cell types. *IL10* signaling pathway, a relevant mechanism in AD multimorbidity across many cell types, was also identified amongst the highest-scoring pathways in our previous study (under the BioCarta denomination *Regulation of hematopoisesis by cytokines*). Our previous study also linked *IL4*-mediated, *GATA3*-mediated mechanisms and 4–1BB-dependent immune responses to ADR multimorbidity. *GATA3*-mediated mechanisms are represented in our dataset by interleukin pathways in the *Cytokine signaling in immune system* category, and, from a cell-type-specific point of view, these processes seem more relevant in AD multimorbidity (despite the fact that *IL1*, *IL4* and *IL13* signaling in particular also contribute to ADR multimorbidity in some cell types). Aside from differences in pathway annotation of genes, this could reflect a more systemic role for these pathways in ADR multimorbidity. However, *4–1BB-dependent immune response* (represented by Toll-like receptor cascades in our dataset) is clearly associated to AR, DR and ADR multimorbidity in a number blood-derived cell types. In all, we believe that our results are complementary to those of our previous study since they focus on the cell-type-specific mechanisms of multimorbidity instead of global (or systemic) ones.

## Conclusions

We designed an *in silico* approach that integrated current public expression and network interaction databases and applied an interactome-based analysis to uncover the cell-type-specific pathophysiological mechanisms of multimorbidity between A, D and R. We observed that interleukin-mediated signaling is present in all multimorbidities involving asthma but not rhinitis, while rhinitis-associated multimorbidities have a strong TLR-mediated component. *IL1* signaling is the only type-2 pathway candidate for AR multimorbidity, found in esophageal epithelium. We also generated a collection of genes potentially linked to cell-type-specific multimorbidity, some of which were not previously associated to any of the diseases. Our results provide a better understanding of the pathophysiological mechanisms triggering ADR multimorbidity, assisting in the design of new mechanistic and clinical studies.

## Supporting information

**S1 Fig. Illustration of the process to calculate cell-type-specific multimorbidity.** This toy example uses a simplified network of the cell type *c*, where we will measure the

multimorbidity score *MS* for diseases *d1* and *d2*. The numbers circled in grey correspond to the numbered steps in the section *Calculating cell-type-specific multimorbidity* of *Methods*. **(A)** Genes associated to *dis1* (6, orange border) are given an initial score of 1, while all other genes are given a score of 0. **(B)** The *NetScore* algorithm scores all genes in the network according to their connectivity to the D-associated genes (blue gradient). Genes in closer proximity to *dis1*-associated genes get higher scores. **(C)** The top-scoring genes are selected (in blue). Disease *dis1* has 13 top-scoring genes ($S^c_{dis1}$). **(D)** Genes associated to *dis2* (5, in orange border) are given an initial score of 1, while all other genes are given a score of 0. **(E)** The *NetScore* algorithm scores all genes according to their connectivity to the *dis2*-associated genes (blue gradient). **(F)** The top-scoring genes are selected (in blue). Disease *dis2* has 47 top-scoring genes ($S^c_{dis2}$). **(G)** There is 1 gene common to both top-scoring sets (in blue). The Multimorbidity Score (*MS*) of the diseases is calculated as the Sorensen-Dice overlap between their top-scoring gene sets. In this case, $MS^c_{dis1,dis2}$ is $(2 \cdot 1) / (6 + 47) = 0.038$. A permutation test over $10^3$ iterations will establish if $MS^c_{dis1,dis2}$ is statistically significant ($P < 0.05$).
(PNG)

**S2 Fig. Illustration of the process to characterize cell-type-specific multimorbidity mechanisms.** This example uses the network of S1 Fig (225 genes). The pathway *P* has a total of annotated 20 genes, of which 9 are in the network (shown in orange border). **(A)** The 13 top-scoring genes for disease *d1* ($S^c_{d1}$; see S1C Fig) are shown in blue, and there are 3 pathway genes within this set. Thus, the perturbation score $PS^c_{d1,P}$ is $(9/20) / (13/225) = 7.79$. For the sake of the example, we will assume that this value is significantly larger than random expectation ($P < 0.05$). **(B)** The 47 top-scoring genes for disease *d2* ($S^c_{d2}$; see S1F Fig) are shown in blue. There are 7 pathway genes within the $S^c_{d2}$ set. Thus, the perturbation score $PS^c_{d2,P}$ is $(9/20) / (47/225) = 2.15$. For the sake of the example, we will assume that this value is significantly larger than random expectation as well ($P < 0.05$). Consequently, because pathway *P* is significantly associated to (or perturbed by) diseases *d1* and *d2*, we assume that it is part of the mechanism of multimorbidity between *dis1* and *dis2* in cell type c.
(PNG)

**S1 Table. Association between Reactome pathways and BioCarta pathways.** Only significant associations are shown. LOR: Log Odds Ratio.
(XLS)

**S2 Table. List of cell-type-specific genes.** This table contains: 1) the database sources of diease-associated genes; 2) the complete list of cell types and tissues (including those without disease-associated genes, discarded in this study); 3) the list of all cell-type-specific genes.
(XLS)

**S3 Table. Fraction of disease-associated genes in each cell type.** Statistical significance was calculated by means of a Fisher's Exact Test.
(XLS)

**S4 Table. Fraction of pathway-associated genes present in each cell type.**
(XLS)

**S5 Table. List of genes associated to each pathway in each cell-type-specific network.**
(XLS)

**S6 Table. The connectivity *Ccp* of the pathways.**
(XLS)

**S7 Table. Summary of Tables 2 and 3.** The column *n diseases* contains the number of diseases (A, D, R) with a significant number of associated genes from Table 2 (values are highlighted in blue gradient). The column *n MS > 0* contains the number of combinations of diseases (AD, AR, DR, ADR) with nonzero *MS* from Table 3 (values are highlighted in red gradient). The column *n MS > 0.50* contains the number of combinations of diseases (AD, AR, DR, ADR) with *MS > 0.50* (also from Table 3, highlighted in red gradient).
(XLS)

**S8 Table. Cellular pathways associated to multimorbidity between asthma, dermatitis and rhinitis.** Red cells: multimorbidity between A and D. Orange cells: multimorbidity between A and R. Light blue cells: multimorbidity between D and R. Dark blue cells: multimorbidity between A, D and R. Only cell types not present in Table 4 in the manuscript are shown.
(XLS)

**S9 Table. Pathways associated to diseases in the cell-type-specific networks.** A: asthma. D: dermatitis. R: rhinitis. Only significant associations (*P* < 0.05) are shown.
(XLS)

**S10 Table. Complete list of candidate genes for multimorbidity.** Colors and dots are as in Tables 5 and 6 in the manuscript. Pathway associations with a grey background mean that the pathway was not associated to the corresponding cell type (see Table 4, S8 Table).
(XLS)

**S11 Table. Comparison of multimorbidity scores.** Scores for AD, AR and DR multimorbidities from Table 5 (30 top-scoring genes) and S10 Table (all genes) are pairwisely compared by means on a Wilcoxo-Mann-Whitney paired test.
(XLS)

**S1 Text. Supplementary Methods.**
(PDF)

## Acknowledgments

We thank Judith García-Aymerich, PhD and Emre Guney, PhD for fruitful discussions.

## Author Contributions

**Conceptualization:** Daniel Aguilar, Stefano Guerra, Josep M. Anto.

**Funding acquisition:** Josep M. Anto.

**Investigation:** Daniel Aguilar.

**Methodology:** Daniel Aguilar, Nathanael Lemonnier, Baldo Oliva.

**Software:** Daniel Aguilar.

**Supervision:** Daniel Aguilar, Stefano Guerra, Jean Bousquet, Josep M. Anto.

**Writing – original draft:** Daniel Aguilar.

**Writing – review & editing:** Daniel Aguilar, Nathanael Lemonnier, Gerard H. Koppelman, Erik Melén, Baldo Oliva, Mariona Pinart, Stefano Guerra, Jean Bousquet, Josep M. Anto.

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
