## [Decision Letter · Decision Letter 0]

29 Aug 2019

PONE-D-19-18278

Understanding allergic multimorbidity within the non-eosinophilic interactome

PLOS ONE

Dear Dr Daniel Aguilar,

Thank you for submitting your manuscript to PLOS ONE. After careful consideration, we feel that it has merit but does not fully meet PLOS ONE’s publication criteria as it currently stands. Therefore, we invite you to submit a revised version of the manuscript that addresses the points raised during the review process.

We would appreciate receiving your revised manuscript by September 27. To enhance the reproducibility of your results, we recommend that if applicable you deposit your laboratory protocols in protocols.io, where a protocol can be assigned its own identifier (DOI) such that it can be cited independently in the future. For instructions see: http://journals.plos.org/plosone/s/submission-guidelines#loc-laboratory-protocols

We look forward to receiving your revised manuscript.

Kind regards,

Davor Plavec

Academic Editor

PLOS ONE

Journal Requirements:

'This work was supported by Mechanisms of the Development of ALLergy (MeDALL), a collaborative project done within the EU under the Health Cooperation Work Programme of the Seventh Framework programme (grant agreement number 261357). EM is supported by grants from the European Research Council (n° 757919) and the Swedish Research Council. NL is a recipient of a postdoctoral fellowship from the French National Research Agency in the framework of the "Investissements d’avenir" program (ANR-15-IDEX-02). The funders had no role in study design, data collection and analysis, decision to publish, or preparation of the manuscript.'

We note that one or more of the authors are employed by a commercial company:6AM Data Mining.

Additional Editor Comments (if provided):

As the Reviewer #1 has serious doubts about the manuscript a major revision or a rebuttal of arguments presented in his review and revision in accordance to the Reviewer #2 should be submitted for additional assessment by the Editor and reviewers before a decision about the manuscript could be made.

Reviewers' comments:

Reviewer's Responses to Questions

**Comments to the Author**

1. Is the manuscript technically sound, and do the data support the conclusions?

Reviewer #1: Partly

Reviewer #2: Yes

2. Has the statistical analysis been performed appropriately and rigorously? 

Reviewer #1: Yes

Reviewer #2: Yes

3. Have the authors made all data underlying the findings in their manuscript fully available?

Reviewer #1: Yes

Reviewer #2: Yes

4. Is the manuscript presented in an intelligible fashion and written in standard English?

Reviewer #1: No

Reviewer #2: Yes

5. Review Comments to the Author

Reviewer #1: General remark

This paper reports on an extensive interactome analysis of three disease known to be clinically associated as co-morbidities, asthma, rhinitis and dermatitis. It follows on from their previous paper in this journal but now focuses on phenotypes of disease that distinctly do not involve eosinophils.

Specific comments

1. This is a study that has used a complex systems biology to integrate data. My main general comment/critique of this paper is that it makes very difficult reading, and makes an assumption that readers will understand the methods and how the data were interpreted. This may well be the case fo a systems biology journal but not for a journal like PLOS which is read widely. I fear that not many people without in-depth knowledge of the methods applied will understand it. I have given some examples below but the whole paper could do with a rewrite to make it more friendly. For example, a standard cell biologist will be puzzled to see that 14 cell types are major components in asthma, rhinitis and dermatitis in 15 distinct tissue sites. It is difficult to comprehend how any tissue outside the lung, the nose and skin (e.g. adipose tissue) is related to any of these diseases.

2. What do the authors conclude is a major novelty that arises out of their analysis? After so much analysis the conclusions seem very bland: the link between asthma and dermatitis through IL-4 and IL-13 (hardly a novel thing) and TLR-mediated IL1 signaling. No proposal for how the genes previously not associated with these diseases could be involved or targeted.

3. The key obstacle which took me a while to work around is that the authors state that the data reflects only expression levels in healthy individuals yet the paper is making conclusions about three diseases!

4. The authors do not provide any no steer for taking the interactome, which is a hypothesis-setting stage, towards definitive proof. One could argue that the only way to do this is to perturb the pathways that are shared across the three diseases.

5. I found it very annoying to stop and think about what terms like cell type T mean. What does disease d (line 107) mean? I am assuming it means disease-associated but I can’t see the value of writing it like this. Why not just say disease-associated. When reading a text, it is annoying to have to go back to the initial definition to see what it means.

1. Cell type-specific gene expression: I am intrigued to see that only samples not subjected to any treatments were considered. Why? An explanation is needed in the methods or, if more lengthy, in the discussion. Also, why discard whole blood derived data if they are controlled for cell types? I am sure the authors are aware of methods to correct for cell types in a sample containing a mix of cells.

2. The sentence “Being reactome a hierarchical collection of pathways..” is not clear at all.

3. Results section, line 285: please state from which databases the disease-gene associations were obtained.

4. Table 1: the legend needs clarification. The full circles distinguish associations characterised by GWAS ad by other methods. What about associations made by both GWAS and other methods? It would be useful to see a column with the number of methods that show the association.

5. I am intrigued to see that more gene-disease associations were shared between asthma and dermatitis than between asthma and rhinitis. What does that say about the one airway concept that implies strong links between the upper and lower airways?

6. Line: 301: We are told that the complete interactome contained 15.332 genes, yet Table 1 only shows a fraction of these as being associated with the three diseases. Does this mean that the other genes (the majority) are NOT associated with the three diseases studied in this paper? If this is correct, then a sentence should be added to explain that we are seeing only a minority of genes associated with the three diseases.

7. Line 303: We are told that there were 62 cell types. Can we see the list of these cell types or, if Table 2 is the full list, please say so in the text?

8. Line: 303: Why and how were cells classified into 15 distinct tissues? What tissues are you talking about (the list?)

9. Table 2 legend: when you say that “the number of genes is significantly high”, high compared to what? Is the adjusted p value FDR adjusted?

10. Line 320: when you say that 519 pathways were available, do you mean that there was evidence for 519 pathways detected in the datasets you examined?

11. Table 3 shows tissues that have no connection with the three diseases. For example, if we take the first tissue type (adipose tissue from abdomen and thigh, please explain how these data are linked to one of the three diseases?

12. I am struggling to understand in Table 4 how Erythrocyte pathways are associated with asthma and rhinitis in respect of the adaptive immune system.

Minor comments:

1. The paper would benefit from proof reading by a native English speaker as there are several minor mistakes in English and punctuation. E.g. abstract: in the methods section error in English “…multimorbidity mechanisms in distinct cell types WERE characterised…., use of the word “coordinately”instead of “in a coordinated manner” (line 60), repetition of “can be shared” (line 75 etc.

Reviewer #2: The authors investigated mechanisms explaining the multimorbidity between asthma, dermatitis and rhinitis and their specificity in distinct cell types by means of an interactome-based analysis of expression data. The authors observed differential roles for cytokine signaling, TLR-mediated signaling and distinct metabolic pathways across distinct cell types. This paper is a continuation of their previous study, where they explored multimorbidity between asthma, dermatitis and rhinitis at whole organism level by investigating patterns of network connectivity between cellular networks.

The paper is well written and data is clearly presented. Data in tables are well presented but figures are not clear. I recommend uploading higher resolution images. The discussion does a nice job of putting the results into context of previous studies. However, it would be good to discuss also other genes for which a significant link has been shown. For example, it would be interesting if you could discuss what is known about PLA267 gene so far in the context of these multimorbidities.

6. PLOS authors have the option to publish the peer review history of their article (what does this mean?). If published, this will include your full peer review and any attached files.

Reviewer #1: No

Reviewer #2: No

---

## [Author Response · Author response to Decision Letter 0]

27 Sep 2019

Response to the Reviewers

When thank the reviewers for their deep review of our manuscript and their useful criticisms and recommendations. We have done our best to have all comments carefully considered and approached.

Reviewer #1

This paper reports on an extensive interactome analysis of three disease known to be clinically associated as co-morbidities, asthma, rhinitis and dermatitis. It follows on from their previous paper in this journal but now focuses on phenotypes of disease that distinctly do not involve eosinophils.

Specific comments

COMMENT #1-A. This is a study that has used a complex systems biology to integrate data. My main general comment/critique of this paper is that it makes very difficult reading, and makes an assumption that readers will understand the methods and how the data were interpreted. This may well be the case of a systems biology journal but not for a journal like PLOS which is read widely. I fear that not many people without in-depth knowledge of the methods applied will understand it. I have given some examples below but the whole paper could do with a rewrite to make it more friendly.

AUTHORS' ANSWER: It is true that our methodology is highly technical and may look abstruse to readers unfamiliar with bioinformatics who are, nonetheless, interested in our study. In order to make the text more friendly to non-specialists, we perused the manuscript (and the Methods section in particular) assisted by the expertise of Dr Jean Bousquet (Editor-in-Chief of Clinical Translational Allergy, and member of the editorial board of a number of journals in the fields of allergy and immunology [https://ctajournal.biomedcentral.com/about/editorial-board/jean-bousquet]). We moved the more technical parts of Methods to a supplementary file (S1 Text) while keeping in the main manuscript a more plain description of the methodology, hopefully easier to follow by non-specialists. For the sake of accuracy, we had to keep some technicalities in the descriptions (e.g. some abbreviations).

Our journal choice followed the acceptance and publication of our previous paper [Aguilar, 2017] which was also an in silico method of similar complexity. We believe that this is not an insolvable problem in a journal such as PLOS ONE, where in-silico-only studies are routinely published, describing complex computational procedures based on machine learning, neuronal systems, prediction software or algorithmics [Kulikovskikh, 2019; Parashar, 2019; Mirabello, 2019; Ingrossom 2019; Gearing, 2019; Shtar, 2019; just to name a recent few].

COMMENT #1-B. For example, a standard cell biologist will be puzzled to see that 14 cell types are major components in asthma, rhinitis and dermatitis in 15 distinct tissue sites. It is difficult to comprehend how any tissue outside the lung, the nose and skin (e.g. adipose tissue) is related to any of these diseases.

AUTHORS' ANSWER: We agree with the reviewer. The involvement of 14 cell types in 15 tissues may sound puzzling to a conventional reader. However this is part of the novelty and an accepted possibility in the systems medicine. A, D and R are complex systemic diseases affecting a wide range of bodily systems through inflammatory-related processes.

Our aim with this study was to systematically assess the involvement of all available cell types in our dataset in the mechanisms of ADR multimorbidity (instead of focusing only on those for which evidence had been previously reported). We believe that by doing so we could potentially suggest new mechanisms or predict the clinical expression of multimorbidity in unexpected novel locations, providing researchers with new insights and directions to explore the molecular nature of ADR multimorbidity.

In this respect, Table 3 does not show the relationship between a disease and a cell type, but the degree with which multimorbidity is manifested in a given cell type (as MS score, ranging from 0 to 1). There are 14 cell types where MS is > 0, suggesting some degree of involvement. However, this does not mean that all these cell types are "major components" of multimorbidity. For instance, kidney epithelium has a score of 0.11, which implies a non-zero but nonetheless low impact of ADR multimorbidity in this cell type, owing probably to its epithelial nature (which is shared with other epithelia where multimorbidity has a stronger manifestation, such as the esophageal epithelium). We rewrote the Quantification of cell-type-specific multimorbidity section in Results to acknowledge this fact (line 286):

Inspection of Table 3 shows 14 cell types associated to ADR multimorbidity because their MS value is > 0 for all combinations of the three diseases (the strength of the association given by the MS value, ranging from 0 to 1).

Re to the particular example of the adipose tissue, there is evidence linking it to inflammation-related conditions through low-grade inflammatory processes [Ouchi, 2011; Greenberg, 2006; Karczewski, 2018], and excess body mass has been linked to the risk of development of asthmatic symptoms [Leiria, 2015; Muc, 2016]. This link has been studied mostly for asthma [Backer, 2016], although there is also evidence of dermatitis-related diseases [Nagel, 2009].

COMMENT #2. What do the authors conclude is a major novelty that arises out of their analysis? After so much analysis the conclusions seem very bland: the link between asthma and dermatitis through IL-4 and IL-13 (hardly a novel thing) and TLR-mediated IL1 signaling. No proposal for how the genes previously not associated with these diseases could be involved or targeted.

AUTHORS' ANSWER: We think that the novelty of our study is two-fold:

First, this is the first study to our knowledge to systematically assess the involvement of many different cells and tissues on the development of ADR multimorbidity. So far, the study of these conditions has been focused in a few cell types (such as eosinophils). To our knowledge, this is the first time that a cell-type-wide landscape of allergic multimorbidity is produced, and our results show that non-eosinophilic cell types have a role in the manifestation of multimorbidity.

Second, we identified specific molecular mechanisms for multimorbidity depending whether rhinitis is present or not: while interleukin-mediated signaling seems to be in the basis of all asthma-involving multimorbidities, the role of TLR-mediated signaling is largely absent when rhinitis is not present as one of the multimorbid diseases. We rephrased the Conclusions to highlight this fact (line 598):

We observed that interleukin-mediated signaling is present in all multimorbidities involving asthma but not rhinitis, while rhinitis-associated multimorbidities have a strong TLR-mediated component.

As to novel gene candidates, Tables 5, 6 and S10 provide lists of candidate genes potentially associated to multimorbidity. Table 6, in particular, provides a list of candidate genes not yet associated to any of the diseases, which makes them particularly interesting subjects of study. However, owing to the large number of potential candidates, we only briefly described a few of the top-scoring ones (namely, IL1RL1, IL13, IL22RA1 and IL20RA). Following the reviewers' suggestions, we expanded the Cell-type-specific candidate genes section in Discussion to include PLA2G7 (the only non-cytokine-related gene among the top 10 in Table 5) and BCHE and proteoglycans NCAN and CSPG5 from Table 6.

COMMENT #3. The key obstacle which took me a while to work around is that the authors state that the data reflects only expression levels in healthy individuals yet the paper is making conclusions about three diseases!

AUTHORS' ANSWER: We agree with the reviewer: it seems hardly possible to infer anything about the mechanisms of diseases by studying the gene expression in healthy subjects. However, this is a standard procedure in interactome-based in silico studies of disease, which face unavoidable limitations in data availability. In our case, those limitations were:

1) To the best of our knowledge, there is not a single expression data study for ADR across multiple tissues/cell types (we discarded merging expression studies from different origins because of the unsurmountable level of data noise it would have added due to technical variation).

2) We lack interactomic data for individuals with ADR (and this is true for many other diseases: the "diseased" interactome is largely undescribed).

Thus, our methodology was designed to convey the disease-related information not in the gene expression levels or the interactome, but in the disease-associated genes. The result is that our study measures how disease (i.e. malfunctioning genes) perturbs the normal (i.e. healthy) cellular mechanisms, represented by the cell-type-specific interactome.

This methodology (and variations thereof) has been extensively used in in silico studies of disease, with remarkable results [Goh, 2007; Vidal, 2011; Bashir, 2014; Kitsak, 2016; Huttlin, 2017]. In a recent review, Sonawane et al. examined how the “healthy” interactome can be mined for the localization of the disease perturbation, better disease sub-type classifications, and better targets for drug development [Sonawane, 2019]. Furthermore, studies and software tools predicting novel disease-associated genes rely on similar methodologies [Guney 2014; Ghiassian, 2015; Huttlin, 2017; just to name a few]. 

A statistical comparison of our results to those obtained from the analysis of gene expression and the interactome data of diseased individuals would have been really interesting. However, the above-mentioned limitations make it impossible. This is why in the Discussion section we compared our findings to those found in literature through a case-by-case manual revision of previous studies.

COMMENT #4. The authors do not provide any no steer for taking the interactome, which is a hypothesis-setting stage, towards definitive proof. One could argue that the only way to do this is to perturb the pathways that are shared across the three diseases.

AUTHORS' ANSWER: Certainly, our objective was to "perturb" the pathways and statistically measure the degree to which those perturbations could lead to multimorbidity in distinct cell types (see COMMENT #3). The lack of an experimental stage in fully computational studies like ours makes them more oriented towards setting new hypothesis and guiding new experiments (see references in COMMENT #1-A). Consequently, our results are predictions to guide further experimental research towards the discovery of new disease-related genes, as has long been one of the main tasks of Bioinformatics [see Yo, 2008; Kann, 2010; Ferrero, 2017; van Dam, 2018 and references therein]. Throughout in the Discussion we highlighted how our predictions agreed with experimental evidences from literature. We have now emphasized the hypothesis-setting role of out study with the following sentence in Strengths section in the Discussion (line 388):

The findings of this in silico study are hypothesis-generating and are intended to guide new experiments on cell-type-specific allergic multimorbidity. Consequently, they should be confirmed by proper mechanistic and genetic studies.

COMMENT #5. I found it very annoying to stop and think about what terms like cell type T mean. What does disease d (line 107) mean? I am assuming it means disease-associated but I can’t see the value of writing it like this. Why not just say disease-associated. When reading a text, it is annoying to have to go back to the initial definition to see what it means.

AUTHORS' ANSWER: Line 107 said: "Genes associated to a disease d (any of A, D or R) will be hereinafter referred to as d-associated genes." Hence, a d-associated gene is a gene associated to A, D or R. Similarly, line 148 said: "Genes specific to a cell type T (any of our cell types of interest) will be hereinafter referred to as T-specific genes".

We discarded the term "disease-associated gene" because it was too ambiguous for an accurate description of some parts of the methodology (what disease would it be referring to?). For the same reasons, we discarded using "cell-type-specific network". Furthermore, the formulae that we employ in the Methods section use those same abbreviations. However, it is true that the term "cell type T" can be confusing, since T cells are a major cell type associated to immune system. This is why we changed it to "cell type c". Also, we rewrote the manuscript to ensure that most of those abbreviations are mostly restricted to the supplementary Methods (S1 Text) only, keeping them down to a minimum in the main text.

COMMENT #6-A. Cell type-specific gene expression: I am intrigued to see that only samples not subjected to any treatments were considered. Why? An explanation is needed in the methods or, if more lengthy, in the discussion. 

AUTHORS' ANSWER: For consistency, it is generally not advisable to combine expression data from healthy individuals with data from patients undergoing a treatment or exposed to some environmental factor. The original expression dataset included data for individuals after alcohol consumption, after sugar consumption, exposed to UV, treated with DMSO, treated with TFG-b1, etc. Aside from the fact that all these treatments and exposures are unrelated to the diseases under study (except maybe for TFG-b1), mixing them with data from healthy individuals would have added noise to the gene expression levels without adding any benefit that we can see.

We added a clarifying sentence in S1 Text (supplementary Methods):

In order to maximize consistency and avoid noise in the gene expression levels, only adult human samples and cell types not subjected to any treatments (e.g. treated with DMSO) neither exposed to particular environmental factors (e.g. tobacco smoke, UV) were considered. 

COMMENT #6-B. Also, why discard whole blood derived data if they are controlled for cell types? I am sure the authors are aware of methods to correct for cell types in a sample containing a mix of cells.

AUTHORS' ANSWER: Yes, there are methods to separate individual cell-types in heterogeneous gene expression data but most of them perform deconvolution-based enrichment analysis (CIBERSORT, X-CELL, LinSeed). In our case, identifying the relative frequencies of cell types within the whole blood "tissue" would have not helped, since we need to know the actual expression levels of all the genes for every cell type and, to our knowledge, no software tool performs this. Furthermore, even if such a tool existed, it would simply provide a statistical estimation of gene expression levels, and we did not wish to draw biological predictions from data which was, in turn, a prediction.

However, most of the major cell types typically associated to whole blood (neutrophils, monocytes/macrophages, lymphocytes, erythrocytes) were already present individually in the expression study. The only exception was eosinophils, but, as we stated in the discussion, its role in ADR multimorbidity has been studied extensively, so we chose to focus on other lesser studied cell types.

COMMENT #7. The sentence “Being reactome a hierarchical collection of pathways..” is not clear at all.

AUTHORS' ANSWER: Rewritten to (line 138):

Reactome is a collection of pathways built in a hierarchical manner, where larger pathways are subdivided into smaller pathways with more specific functionalities.

COMMENT #8. Results section, line 285: please state from which databases the disease-gene associations were obtained.

AUTHORS' ANSWER: It's in the Methods section > Data sources > Gene-disease associations. We rewrote the text to point the reader to that section (line 227):

The complete list of genes is shown in Table 1 (see Table S2 and Gene-disease associations in the Methods section for data sources). 

COMMENT #9. Table 1: the legend needs clarification. The full circles distinguish associations characterised by GWAS ad by other methods. What about associations made by both GWAS and other methods? It would be useful to see a column with the number of methods that show the association.

AUTHORS' ANSWER: We agree, the text of the legend in Table 1 was confusing. It was rewritten and now reads:

Table 1. Gene-disease associations. A: asthma; D: dermatitis; R: rhinitis. Filled circle: all evidences. Empty circle: GWAS-only evidence. Only genes with expression data, present in the interactome and associated to A, D or R are shown. 

Re to the number of distinct methods of characterization, this is a kind of information that is not explicitly provided in the databases that we used (with the exception of PheGenI, which contains only GWAS-based associations). Supplementary Table S2 now shows the database(s) from where the associations were extracted.

COMMENT #10. I am intrigued to see that more gene-disease associations were shared between asthma and dermatitis than between asthma and rhinitis. What does that say about the one airway concept that implies strong links between the upper and lower airways?

AUTHORS' ANSWER: Not all gene-disease associations have the same clinical impact. Particularly in complex diseases, the molecular relationship between an altered gene and the manifestation of a disease can take many forms, which translates into some genes having a larger impact in the disease than others. We are not aware of any kind of metric measuring the clinical impact of a gene-disease association, and sometimes (for instance in GWAS studies) the nature of this molecular relationship can only be assumed. On top of this, different databases have different criteria as to incorporate gene-disease associations. For these reasons, 1) Table 1 provides the gene-disease associations but does not quantify them in terms of clinical impact, and 2) it is entirely possible for asthma and rhinitis to have a stronger molecular relationship (sharing only 8 genes) than asthma and dermatitis (sharing 22 genes).

We partially addressed that problem by analyzing multimorbidity at interactome level instead of gene level. Interactome-based studies have shown that topology in the network can be used as a proxy for clinical impact (the alteration of a more central gene is likely to have a larger impact in the surrounding interactome) [Jeong 2001; Vidal, 2011; Carson, 2015; Park, 2019; just to name a few]. So, our multimorbidity scores can be interpreted as a measure of the impact of a gene in multimorbidity. Tables 5 and 6 (which are top-scoring subsets of Table S10) show that, for the same gene, scores for AR multimorbidity tend to be larger than scores for AD multimorbidity (statistical comparison in newly-added Table S11). This suggests that the molecular relationship between asthma and rhinitis is stronger than the relationship between asthma and dermatitis despite having less shared gene in Table 1.

We have expanded the Candidate multimorbidity genes section in Results to acknowledge this fact (line 331):

Genes in Table 5 show a higher score, on average, for AR than for AD multimorbidity (P = 0.01482; paired Wilcoxon-Mann-Whitney test), implying a more closely-knit biological mechanism for AR than for AD multimorbidity. The same was observed for AD vs DR (P = 1.02·10-3; paired Wilcoxon-Mann-Whitney test) but not for AR vs DR. This observation was also true when comparing scores of the whole set of predicted genes in S10 Table.

We also expanded the Cell-type-specific candidate genes section in Discussion (line 536):

The "United Airways" concept, introduced in the previous section, is also supported by our results: on average, mechanisms mediating between A and R are more closely-knit (represented by higher average scores) than mechanisms mediating A and D, despite the fact that A and D share more disease-associated genes (Table 1).

Lastly, we cannot rule out the presence of a research bias: asthma and dermatitis have been much more studied than rhinitis (a quick PubMed search retrieves 182,260 entries for asthma, 120,033 entries for dermatitis, and 43,007 entries for rhinitis). This suggests a larger number of disease-associated genes known for asthma and dermatitis, which in turns increases the chance of a larger overlap.

COMMENT #11. Line: 301: We are told that the complete interactome contained 15.332 genes, yet Table 1 only shows a fraction of these as being associated with the three diseases. Does this mean that the other genes (the majority) are NOT associated with the three diseases studied in this paper? If this is correct, then a sentence should be added to explain that we are seeing only a minority of genes associated with the three diseases.

AUTHORS' ANSWER: Yes, only a small fraction of the genome is associated to A, D or R. Naturally, we cannot rule out that in the future new genes will be associated to A, D or R (they certainly will) but we reflected the state of the knowledge at the present moment.

Table 1 shows the genes associated to at least one of the diseases. The legend has been rewritten to clarify this fact (see answer to COMMENT #9).

COMMENT #12. Line 303: We are told that there were 62 cell types. Can we see the list of these cell types or, if Table 2 is the full list, please say so in the text?

AUTHORS' ANSWER: Table 2 only shows those cell types where at least one disease-associated gene has been found. So, the last sentence in Table 2 legend has been rewritten as:

For clarity, zero values are represented as blank cells, and cell types without any disease-associated genes are not shown.

Although tissues and cell types are part of the information provided in Table S2, for clarity we have expanded Supplementary Table S2 with the list of tissues and cell types only. In creating this table, we realized that the total number of cell types was 60, not 62 (after the removal of generic cell types peripheral blood leukocytes and peripheral blood mononuclear cells, see Methods). This figure has been corrected in lines 241 and 284 in the text.

COMMENT #13. Line: 303: Why and how were cells classified into 15 distinct tissues? What tissues are you talking about (the list?)

AUTHORS' ANSWER: The classification of cell types into tissues was obtained from the original expression study E-MTAB-62 [https://www.ebi.ac.uk/arrayexpress/experiments/E-MTAB-62/]. Supplementary Table S2 provides the complete list of tissues as well as their corresponding cell types.

COMMENT #14. Table 2 legend: when you say that “the number of genes is significantly high”, high compared to what? Is the adjusted p value FDR adjusted?

AUTHORS' ANSWER: In Table 2, the sentence "the number of genes is significantly high" means that the number of genes is significantly higher than random expectation. Throughout the paper, the term "significant" is always associated to statistical significance. We have rephrased the legend accordingly.

As explained in Methods, we adjusted P-values with the Benjamini-Hochberg method, which controls the False Discovery Rate (FDR) [Haynes, 2013]. We rewrote the first mention of the method to acknowledge this (line 154). Now reads:

[...] adjusted by the Benjamini-Hochberg method for false discovery (FDR) control.

COMMENT #15. Line 320: when you say that 519 pathways were available, do you mean that there was evidence for 519 pathways detected in the datasets you examined?

AUTHORS' ANSWER: No, we meant that 519 were available in the Reactome database after removing overlapping pathways. The line has been rewritten as (line 261):

The number of pathways in Reactome database was 519 after filtering, with an average pairwise overlap of 0.01%.

COMMENT #16. Table 3 shows tissues that have no connection with the three diseases. For example, if we take the first tissue type (adipose tissue from abdomen and thigh, please explain how these data are linked to one of the three diseases?

AUTHORS' ANSWER: This study was not aimed to explore the mechanisms of single diseases, but only the mechanisms of multimorbidity between pairs/trios of diseases (i.e. AD, AR, DR, ADR). As explained in Methods, two diseases were considered to display multimorbidity in a certain cell type if (1) they both were manifested individually in the cell type, and (2) they perturbed pathways that overlapped (implying that the manifestation of one disease can be “transmitted” as a perturbation through the pathway and cause the manifestation of the other, hence the multimorbidity). The MS score in Table 3 numerically measures that overlap. We have rewritten the "Quantification of cell-type-specific multimorbidity" paragraph in Results to clarify this (see COMMENT #1-B).

In Table 3, empty cells mean that no perturbed pathways have been found overlapping between diseases (hence, MS = 0). For instance, for the first cell type "adipose tissue from abdomen and thigh", we found a possible mechanism for multimorbidity between A and D (described in Table 4 as related to interleukin-4, -10, and -13). However, we didn't identify any mechanism for multimorbidity between A-R, D-R or A-D-R for this cell type. In the answer to COMMENT ##1-B we provided some references for studies of the association between these diseases and adipose tissue.

COMMENT #17. I am struggling to understand in Table 4 how Erythrocyte pathways are associated with asthma and rhinitis in respect of the adaptive immune system.

AUTHORS' ANSWER: This is due to a limitation in our study, related to the very definition of what a pathway is. It is known that nonnucleated cells (such as the erythrocyte) express little or no MHC. However, according to Reactome database two asthma-associated genes (CD14, TLR1) and three rhinitis-associated genes (CD14, TLR1, TLR2) are associated to the pathway "Antigen processing-Cross presentation", and we found them specifically expressed in erythorcytes. The reason for this is probably the broad definition of what constitutes the "Antigen processing-Cross presentation" pathway in Reactome database, which includes genes belonging to other pathways not related to the adaptative immune system. In particular, TLR1, TL2 and CD14 are also present in a number of Toll-like receptor cascades, found to be active in erythrocytes [Anderson, 2018; Hotz, 2018]. Since the limits of what can be and cannot be considered as a part of a pathway is always open to discussion (with the possible exception of some classical pathways such as the Krebs Cycle), we chose to use the pathways as they were defined in Reactome database, removing only those with a large overlap (> 50% of genes in common). We have modified the Discussion to acknowledge this fact (line 474):

We have to point out, however, that the definition of a pathway (as a functionally annotated gene set) should be taken into account when analyzing those results. For instance, the pathway Antigen Processing and Cross-Presentation is associated to AR multimorbidity in erythrocytes (Table 4) contradicts evidence on MHC presence in human nonnucleated cells [93] because of the definition of the pathway in Reactome database, that includes genes also annotated in TLR-mediated pathways.

Minor comments:

COMMENT #18. The paper would benefit from proof reading by a native English speaker as there are several minor mistakes in English and punctuation. E.g. abstract: in the methods section error in English “…multimorbidity mechanisms in distinct cell types WERE characterised…., use of the word “coordinately”instead of “in a coordinated manner” (line 60), repetition of “can be shared” (line 75 etc.

AUTHORS' ANSWER: We have proof-read all texts to correct any mistakes. However, according to https://www.dictionary.com (which is based on the Random House Unabridged Dictionary), the adverb "coordinately" is correct.

Reviewer #2

COMMENT #1. The authors investigated mechanisms explaining the multimorbidity between asthma, dermatitis and rhinitis and their specificity in distinct cell types by means of an interactome-based analysis of expression data. The authors observed differential roles for cytokine signaling, TLR-mediated signaling and distinct metabolic pathways across distinct cell types. This paper is a continuation of their previous study, where they explored multimorbidity between asthma, dermatitis and rhinitis at whole organism level by investigating patterns of network connectivity between cellular networks.

The paper is well written and data is clearly presented. Data in tables are well presented but figures are not clear. I recommend uploading higher resolution images. The discussion does a nice job of putting the results into context of previous studies. However, it would be good to discuss also other genes for which a significant link has been shown. For example, it would be interesting if you could discuss what is known about PLA267 gene so far in the context of these multimorbidities.

AUTHORS' ANSWER: Images were checked for resolution using the PACE software, as requested by the journal’s guidelines. All of them passed the quality control. However, the pdf generated for revision contains lower-quality versions of the figures (in order to keep the file size within reasonable limits). A better-quality version of the figure can be downloaded by clicking on the link on the upper right corner of each page.

Because the number of candidate genes was rather long, we chose to focus on those with scarcer previous evidence of association with the diseases under study. However, at the reviewer's request, we have extended the Cell-type-specific candidate genes section in the Discussion to include PLA2G7. We have also extended the section by commenting on the cell type specificity of IL4 and IL5 and by pointing out the higher scores of genes associated with AR multimorbidity when compared to AD multimorbidity, linking this fact to the "United Airways" concept that established a strong mechanistic connection between asthma and rhinitis. The additional text reads (line 530 onwards):

The only non-cytokine-related gene in the top 10 positions of Table 5 is the PLA2G7 gene, which controls inflammation though the inactivation of platelet-activating factor (PAF), a potent phospholipid-derived mediator of inflammation that is secreted by many immune cells and controls vascular permeability. Although no study associating PLA2G7 to ADR multimorbidity exists, it is a strong candidate if we take into account the wide range of actions of PAF (considered a universal biological regulator [111]) that in turn associates PLA2G7 to a number of inflammatory conditions other than A, D or R [112-114]. The "United Airways" concept, introduced in the previous section, is also supported by our results: on average, mechanisms mediating between A and R are more closely-knit (represented by higher average scores) than mechanisms mediating A and D, although A and D share more disease-associated genes (Table 1).

Some of the highest-scoring candidate genes were not even associated to any of the diseases of interest (Table 6) illustrating the potential of our approach to characterize yet-undescribed molecular mechanisms of multimorbidity. For instance, we predict interleukin receptors IL22RA1 and IL20RA to play an important role in the ADR multimorbidity in the esophageal epithelium. To date, IL22RA1 had been only associated to inflammatory responses in airway epithelia by genetic studies, and IL20RA to psoriasis [115, 116]. Also, the functional nature of genes in Table 6 is also much more diverse than that of Table 5. This is strongly suggestive of a research bias towards already-known cytokine-related mediators when it comes to the study of these allergic diseases, overlooking other functional groups. For instance, the second high-scoring gene in Table 6 is butyrylcholinesterase BCHE, a poorly-studied detoxifying enzyme that has been proposed as a marker to identify and prognose systemic inflammation [117, 118] and that has only marginally associated to allergic diseases. BCHE is highlighted by our method as a mediator in ADR multimorbidity in skin. Table 6 also shows that the role of proteoglycans seems to be restricted to AR multimorbidity only through neurocan (NCAN) and chondroitin sulfate proteoglycan 5 (CSPG5). Although proteoglycans are known to influence the remodelling of nasal mucosa in R [119], no evidence exists linking them to allergic multimorbidity. However, our results indicate that TLRs are characteristically associated to multimorbidity involving R (Table 4), so there may be an interesting link between TLRs and proteoglycans in relation to AR multimorbidity, since it is known that chondroitin sulfate proteoglycans have the ability to bind TLRs and activating macrophages [120].

References

Aguilar D, Pinart M, Koppelman GH, Saeys Y, Nawijn MC, Postma DS et al. Computational analysis of multimorbidity between asthma, eczema and rhinitis. PLoS One. 2017 Jun 9;12(6):e0179125

Anderson HL, Brodsky IE, Mangalmurti NS. The Evolving Erythrocyte: Red Blood Cells as Modulators of Innate Immunity. J Immunol. 2018 Sep 1;201(5):1343-1351.

Backer V, Baines KJ, Powell H, Porsbjerg C, Gibson PG. Increased asthma and adipose tissue inflammatory gene expression with obesity and Inuit migration to a western country. Respir Med. 2016 Feb;111:8-15.

Barshir R, Shwartz O, Smoly IY, Yeger-Lotem E. Comparative analysis of human tissue interactomes reveals factors leading to tissue-specific manifestation of hereditary diseases. PLoS Comput Biol. 2014 Jun 12;10(6):e1003632

Carson MB, Lu H. Network-based prediction and knowledge mining of disease genes. BMC Med Genomics. 2015;8 Suppl 2:S9.

Ferrero E, Dunham I, Sanseau P. In silico prediction of novel therapeutic targets using gene-disease association data. J Transl Med. 2017 Aug 29;15(1):182.

Gearing LJ, Cumming HE, Chapman R, Finkel AM, Woodhouse IB, Luu K, Gould JA, Forster SC, Hertzog PJ. CiiiDER: A tool for predicting and analysing transcription factor binding sites. PLoS One. 2019 Sep 4;14(9):e0215495

Ghiassian SD, Menche J, Barabási AL. A DIseAse MOdule Detection (DIAMOnD) algorithm derived from a systematic analysis of connectivity patterns of disease proteins in the human interactome. PLoS Comput Biol. 2015 Apr 8;11(4):e1004120

Goh KI, Cusick ME, Valle D, Childs B, Vidal M, Barabási AL. The human disease network. Proc Natl Acad Sci U S A. 2007 May 22;104(21):8685-90

Greenberg AS, Obin MS. Obesity and the role of adipose tissue in inflammation and metabolism. Am J Clin Nutr. 2006 Feb;83(2):461S-465S

Guney E, Oliva B. Analysis of the robustness of network-based disease-gene prioritization methods reveals redundancy in the human interactome and functional diversity of disease-genes. PLoS One. 2014 Apr 14;9(4):e94686

Gupta P, Vijayan VK, Bansal SK. Changes in protein profile of erythrocyte membrane in bronchial asthma. J Asthma. 2012 Mar;49(2):129-33

Hotz MJ, Qing D, Shashaty MGS, Zhang P, Faust H, Sondheimer N, Rivella S, Worthen GS, Mangalmurti NS. Red Blood Cells Homeostatically Bind Mitochondrial DNA through TLR9 to Maintain Quiescence and to Prevent Lung Injury. Am J Respir Crit Care Med. 2018 Feb 15;197(4):470-480

Huttlin EL, Bruckner RJ, Paulo JA, Cannon JR, Ting L, Baltier K, Colby G, Gebreab F, Gygi MP, Parzen H, Szpyt J, Tam S, Zarraga G, Pontano-Vaites L, Swarup S, White AE, Schweppe DK, Rad R, Erickson BK, Obar RA, Guruharsha KG, Li K,Artavanis-Tsakonas S, Gygi SP, Harper JW. Architecture of the human interactome defines protein communities and disease networks. Nature. 2017 May 25;545(7655):505-509

Ingrosso A, Abbott LF. Training dynamically balanced excitatory-inhibitory networks. PLoS One. 2019 Aug 8;14(8):e0220547

Jeong H, Mason SP, Barabási AL, Oltvai ZN. Lethality and centrality in protein networks. Nature. 2001 May 3;411(6833):41-2

Kann MG. Advances in translational bioinformatics: computational approaches for the hunting of disease genes. Brief Bioinform. 2010 Jan;11(1):96-110

Karczewski J, Śledzińska E, Baturo A, Jończyk I, Maleszko A, Samborski P, Begier-Krasińska B, Dobrowolska A. Obesity and inflammation. Eur Cytokine Netw. 2018 Sep 1;29(3):83-94

Kitsak M, Sharma A, Menche J, Guney E, Ghiassian SD, Loscalzo J, Barabási AL. Tissue Specificity of Human Disease Module. Sci Rep. 2016 Oct 17;6:35241

Kolmykov SK, Kondrakhin YV, Yevshin IS, Sharipov RN, Ryabova AS, Kolpakov FA. Population size estimation for quality control of ChIP-Seq datasets. PLoS One. 2019 Aug 29;14(8):e0221760.

Kulikovskikh I, Prokhorov S, Lipić T, Legović T, Šmuc T. BioGD: Bio-inspired robust gradient descent. PLoS One. 2019 Jul 5;14(7):e0219004

Leiria LO, Martins MA, Saad MJ. Obesity and asthma: beyond T(H)2 inflammation. Metabolism. 2015 Feb;64(2):172-81

Marinho PRD, Silva RB, Bourguignon M, Cordeiro GM, Nadarajah S. AdequacyModel: An R package for probability distributions and general purpose optimization. PLoS One. 2019 Aug 26;14(8):e0221487

Mirabello C, Wallner B. rawMSA: End-to-end Deep Learning using raw Multiple Sequence Alignments. PLoS One. 2019 Aug 15;14(8):e0220182

Muc M, Mota-Pinto A, Padez C. Association between obesity and asthma – epidemiology, pathophysiology and clinical profile. Nutr Res Rev. 2016 Dec;29(2):194-201

Nagel G, Koenig W, Rapp K, Wabitsch M, Zoellner I, Weiland SK. Associations of adipokines with asthma, rhinoconjunctivitis, and eczema in German schoolchildren. Pediatr Allergy Immunol. 2009 Feb;20(1):81-8

Ouchi N, Parker JL, Lugus JJ, Walsh K. Adipokines in inflammation and metabolic disease. Nat Rev Immunol. 2011 Feb;11(2):85-97

Parashar P, Chen CH, Akbar C, Fu SM, Rawat TS, Pratik S, Butola R, Chen SH, Lin AS. Analytics-statistics mixed training and its fitness to semisupervised manufacturing. PLoS One. 2019 Aug 13;14(8):e0220607

Park J, Hescott BJ, Slonim DK. Pathway centrality in protein interaction networks identifies putative functional mediating pathways in pulmonary disease. Sci Rep. 2019 Apr 10;9(1):5863.

Sharma A, Kitsak M, Cho MH, Ameli A, Zhou X, Jiang Z, Crapo JD, Beaty TH, Menche J, Bakke PS, Santolini M, Silverman EK. Integration of Molecular Interactome and Targeted Interaction Analysis to Identify a COPD Disease Network Module. Sci Rep. 2018 Sep 27;8(1):14439

Shtar G, Rokach L, Shapira B. Detecting drug-drug interactions using artificial neural networks and classic graph similarity measures. PLoS One. 2019 Aug 1;14(8):e0219796.

Sonawane AR, Weiss ST, Glass K, Sharma A. Network Medicine in the Age of Biomedical Big Data. Front Genet. 2019 Apr 11;10:294. 

van Dam S, Võsa U, van der Graaf A, Franke L, de Magalhães JP. Gene co-expression analysis for functional classification and gene-disease predictions. Brief Bioinform. 2018 Jul 20;19(4):575-592

Vidal M, Cusick ME, Barabási AL. Interactome networks and human disease. Cell. 2011 Mar 18;144(6):986-98

Vidal M, Cusick ME, Barabási AL. Interactome networks and human disease. Cell. 2011 Mar 18;144(6):986-98

Yu B. In silico gene discovery. Methods Mol Med. 2008;141:1-22.

Zhou J, Chen L, Liu Z, Sang L, Li Y, Yuan D. Changes in erythrocyte polyunsaturated fatty acids and plasma eicosanoids level in patients with asthma. Lipids Health Dis. 2018 Sep 1;17(1):206

---

## [Decision Letter · Decision Letter 1]

15 Oct 2019

Understanding allergic multimorbidity within the non-eosinophilic interactome

PONE-D-19-18278R1

Dear Dr. Daniel Aguilar,

We are pleased to inform you that your manuscript has been judged scientifically suitable for publication and will be formally accepted for publication once it complies with all outstanding technical requirements.

With kind regards,

Davor Plavec

Academic Editor

PLOS ONE

Additional Editor Comments (optional):

Dear Authors, your submission is accepted for publication in its current form.

Reviewers' comments:

Reviewer's Responses to Questions

**Comments to the Author**

1. If the authors have adequately addressed your comments raised in a previous round of review and you feel that this manuscript is now acceptable for publication, you may indicate that here to bypass the “Comments to the Author” section, enter your conflict of interest statement in the “Confidential to Editor” section, and submit your "Accept" recommendation.

Reviewer #1: All comments have been addressed

Reviewer #2: All comments have been addressed

2. Is the manuscript technically sound, and do the data support the conclusions?

Reviewer #1: Yes

Reviewer #2: Yes

3. Has the statistical analysis been performed appropriately and rigorously? 

Reviewer #1: Yes

Reviewer #2: Yes

4. Have the authors made all data underlying the findings in their manuscript fully available?

Reviewer #1: Yes

Reviewer #2: Yes

5. Is the manuscript presented in an intelligible fashion and written in standard English?

Reviewer #1: Yes

Reviewer #2: Yes

6. Review Comments to the Author

Reviewer #1: This is still not a very easy paper to read for people unfamiliar with the complex methods used but I accept that there is no toom for further simplification.

Reviewer #2: The authors addressed all my previous comments. As suggested, the authors extended the Cell-type-specific candidate genes section in the Discussion to include PLA2G7.

The paper uses very specific methodology that is understood by a smaller number of readers who are experts in the field of interactome analysis. However, the revised article is significantly more understandable to a wider audience of readers and is now written clearly enough to be accessible to non-specialists interested in this particular field.

7. PLOS authors have the option to publish the peer review history of their article (what does this mean?). If published, this will include your full peer review and any attached files.

Reviewer #1: No

Reviewer #2: No

---

## [Editor Report · Acceptance letter]

24 Oct 2019

PONE-D-19-18278R1 

Understanding allergic multimorbidity within the non-eosinophilic interactome 

Dear Dr. Aguilar:

I am pleased to inform you that your manuscript has been deemed suitable for publication in PLOS ONE. Congratulations! Your manuscript is now with our production department. 

With kind regards,

on behalf of

Dr. Davor Plavec 

Academic Editor

PLOS ONE